# YAP regulates cell size and growth dynamics via non-cell autonomous mediators

Douaa Mugahid, Marian Kalocsay, Xili Liu, Jonathan Scott Gruver, Leonid Peshkin, Marc W Kirschner*

Department of Systems Biology, Harvard Medical School, Boston, United States

**Abstract** The Hippo pathway regulates organ size, regeneration, and cell growth by controlling the stability of the transcription factor, YAP (Yorkie in *Drosophila*). When there is tissue damage, YAP is activated allowing the restoration of homeostatic tissue size. The exact signals by which YAP is activated are still not fully understood, but its activation is known to affect both cell size and cell number. Here we used cultured cells to examine the coordinated regulation of cell size and number under the control of YAP. Our experiments in isogenic HEK293 cells reveal that YAP can affect cell size and number by independent circuits. Some of these effects are cell autonomous, such as proliferation, while others are mediated by secreted signals. In particular CYR61, a known secreted YAP target, is a non-cell autonomous mediator of cell survival, while another unidentified secreted factor controls cell size.

## Introduction

Organ and tissue size are tightly regulated; they scale with body size, and in several cases can be regenerated after partial loss (*Penzo-Méndez and Stanger, 2015*). The size and number of cells comprising an organ together determine overall size, and these can compensate for each other (*Fankhauser, 1945*). In some well-studied cases adult organ repair and regeneration involves de-differentiation, followed by proliferation and re-differentiation; while in others, such as skeletal and cardiac muscle, the dedifferentiation-re-differentiation process is skipped, and organs adapt to demand by increasing cell size; or to injury by ECM deposition (fibrosis). The mass of β-cells of the pancreas can increase by augmenting both cell size (hypertrophy) or cell number (hyperplasia) (*Anzi et al., 2018*; *Ernst et al., 2011*). Apoptosis can also play an important role in adult organ size homeostasis. For example, the liver enlarges significantly after phenobarbital treatment, but is restored to its normal size by apoptosis upon drug withdrawal (*Yang and Xu, 2011*). These examples demonstrate that organ size is not simply set during development and passively maintained into adulthood but is actively monitored to fit physiological demands.

The liver offers quite a dramatic example of size control. Although hepatocytes are largely quiescent in adults, the liver can regenerate completely after resection of up to 70% of its original mass (partial hepatectomy). This response is tri-phasic. It starts with an increase in the size (hypertrophy) of the remaining hepatocytes, followed by a period of cell proliferation producing daughter cells of normal size, and is terminated when the original liver mass is restored (*Gilgenkrantz and Collin de l'Hortet, 2018*). If, in the case of a partial hepatectomy, hepatocytes are blocked from proliferation, liver size can still be restored by hypertrophy of the remaining, often polyploid cells (*Diril et al., 2012*). This provides a powerful example of how tissue homeostasis can be maintained by a dynamic combination of cell growth and proliferation.

While the role of cell growth in proliferation-dependent liver size homeostasis has never been explicitly tested, it is tempting to speculate that after a partial hepatectomy quiescent hepatocytes

*For correspondence:
marc@hms.harvard.edu

Competing interests: The authors declare that no competing interests exist.

re-enter the cell cycle only after increasing in size and that perturbations that impair growth under such conditions impair regeneration (*Volarevic et al., 2000*). Because of the interdependence of cell size and cell cycle control it is hard to distinguish whether a genetic perturbation that affects proliferation during regeneration primarily affects the cell cycle or cell growth machinery. Though cultured cells provide almost none of the tissue-level context for studying regeneration, they still robustly control cell size by compensatory changes in growth and division (*Cadart et al., 2018*; *Ginzberg et al., 2018*).

There is increasing evidence that the Hippo pathway plays a significant role in tissue size regulation (*Dong et al., 2007*) as well as regeneration (*Lu et al., 2018*). The pathway is comprised of a cascade of intracellular kinases, activated in response to cell-cell contact. These kinases ultimately phosphorylate and inhibit the downstream transcriptional regulators YAP and TAZ, by causing their sequestration, and often degradation in the cytoplasm (*Zhao et al., 2010*). Upon tissue injury, the upstream kinases become inactive and YAP/TAZ translocates into the nucleus, where it mediates transcriptional changes that facilitate organ regrowth (*Lee et al., 2014*). YAP/TAZ activity also depends on changes in the composition and physical properties of the extracellular environment, which is believed to be independent of Hippo activity (*Aragona et al., 2013*; *Dupont et al., 2011*; *Kim and Gumbiner, 2015*). Activation of YAP increases cell proliferation and inhibits apoptosis in vitro and in vivo (*Dong et al., 2007*; *Xin et al., 2011*; *Zanconato et al., 2015*). While the phenomena have been well characterized, the exact molecular mechanism by which YAP affects these different processes and/or their interdependence remains unclear.

In this report, we investigated the mechanisms by which YAP regulates cell number (through proliferation and death) and cell growth/size in mammalian cultured cells. The term 'cell growth' is commonly used in two very different contexts. It is important to distinguish these processes because 'exponential growth' has meant either an exponential increase of cell mass in each cell cycle or an exponential increase in the number of cells per unit time. In this paper we will use the term, 'cell growth' exclusively to describe changes in average *cell size*, and 'population growth' exclusively to describe changes in the *number of cells*.

Our results suggest that YAP affects both cell size and number through distinct mechanisms. Some of these effects appear to be cell-autonomous, but notably we find strong evidence that others are driven by diffusible, non-cell autonomous factors. In particular we show that CYR61, one of the extracellular-residing proteins that is regulated both on the mRNA and protein level in response to increased YAP activity, has no effect on cell size. However, CYR61 mediates a significant part of the YAP-dependent increase in cell number by limiting apoptosis in high density cultures. We conclude that YAP stimulates the production of a suite of extracellular and cell surface proteins that act in a paracrine and autocrine manner regulating both cell number and size.

## Results

### Overexpression of non-phosphorylatable YAP increases the rate of cell division, cell size, and maximum cell density in HEK293 populations

To study the effect of YAP on cell size and growth kinetics, we generated Dox-inducible, isogenic Fli-pinTrex-HEK293 cells expressing nuclear mCherry (n-mCherry) together with either nuclear GFP (nGFP) or a GFP-tagged phosphosite mutant of YAP (YAP5SA). Unlike wildtype (wt-) YAP, YAP5SA cannot be inhibited by the upstream kinases of the Hippo pathway and therefore, is constitutively active. The fact that the cell lines are isogenic, ensures that any observed changes in size or proliferation rate between the cell lines do not arise from the genomic location of the inserts, but rather to differences in biological activity of the expressed proteins. Conveniently the n-mCherry demarks the nucleus, enabling easy counting of cell number in low magnification images (*Figure 1—figure supplement 1*). The nuclear marker also serves another important function, as an indirect measure of cell size (*Ginzberg et al., 2018*). As shown in *Figure 1—figure supplement 2*, this correlation is very strong when cell size is measured by the highly accurate Quantitative Phase Microscopy (QPM). The Pearson correlation coefficient was ~0.9 when measured under various conditions. With this experimental design, we could conveniently track changes in cell number and cell size in a population over time.

We found that the population growth rate in these cells decreased as cell density increased, and eventually reached zero at high cell density (*Figure 1A*). This is a well-known phenomenon often referred to as density-dependent growth inhibition (*Stoker and Rubin, 1967*) or contact inhibition; and can be described by a logistic growth expression, where the net growth rate of a population is exponential at low density and decreases to a minimum at high density (*Figure 1—figure supplement 3A–C*) (*Verhulst, 1845*). In this phenomenological description, changes in the maximum cell number are denoted by *Ymax,* also known as the population carrying capacity; and the growth rate is described by *k* (with units of reciprocal time).

We initially thought that Ymax would only be affected by the area of the culture vessel (literally the carrying capacity), while k would be a function of the concentration of growth factors in the medium. Indeed, serum concentrations above 1% changed k (*Figure 1—figure supplement 3D*) without changing Ymax (*Figure 1—figure supplement 3E*). However, Ymax was lower at serum concentration <1% suggesting that the area of the culture vessel was not the only determinant of Ymax.

A more complete description should not consider proliferation as the sole determinant of cell number but also include the cell death rate. By measuring cell death directly by YOYO-1, a cell impermeable dye that fluoresces green when bound to dsDNA in dead cells, we could show that cell death rates are low in low-density cultures (*Figure 1—figure supplement 4A,B*). Hence at low density the net population growth is dominated by the cell division rate. At high density, when net population growth is zero, the death rate dramatically increases (*Figure 1—figure supplement 4A*). Thus extra- and intra- cellular signals appear to affect k and Ymax differently, with the former predominantly reflecting cell division rates and the latter, cell death rates.

Nuclear area in low density cultures (<~15,000 cells/well in a 96-well plate) is initially constant at ~120 $\mu m^2$ in control cells, after which it decreased about 33% to ~85 $\mu m^2$ at maximum density (*Figure 1B*). Assuming the nucleus is a perfect sphere, this change in area would correspond to a 50% decrease in volume. However, actual measurements of *cell* volume using Coulter Counter indicate a ~ 20% decrease between low density (25% confluent) and high density (100% confluent) cultures (*Figure 1—figure supplement 4C*). The closer correlation between nuclear area and volume than predicted by a spherical representation of the nucleus/cell is consistent with the strong correlation we observed between dry mass and nuclear *area* (*Figure 1—figure supplement 2A*).

The expression of YAP5SA increases both *k* and *Ymax* compared to controls (*Figure 1C*). Furthermore, at low cell density the area of the nucleus increases 17% in the YAP5SA cells. A change of such magnitude would translate to ~30% increase in volume, which we measured by Coulter Counter to be ~25% larger (*Figure 1—figure supplement 5A*). However, as the number of cells continued to grow, the nuclear area of YAP5SA cells also decreased to a minimum where the difference in size compared with controls was much reduced (~10%; *Figure 1D*). The onset of the exponential decrease was slightly delayed when cells overexpressed YAP5SA, though cultures appear to be of comparable density (*Figure 1E*).

To confirm that the changes in nuclear area reflected changes in biomass and not just changes in volume, we measured protein mass directly in individual cells using a covalent dye specific for lysine groups on the protein, which is a proxy for changes in size (*Kafri et al., 2013*). We found that average protein content per cell was ~30% higher in low-density populations expressing YAP5SA (*Figure 1F,G*) than in nGFP controls. We also observed that cells seeded at four times the density had ~25% less protein (*Figure 1F,G*). These measurements of protein mass reinforced both of the conclusions drawn from the changes in nuclear area: (1) YAP5SA cells are larger than control cells in low density cultures, and (2) cell size decreases at higher cell population density. We ruled out that the changes were merely due to an increase in the fraction of cells in S or G2/M by simultaneously measuring DNA and protein (*Figure 1—figure supplement 5B–C*). The latter findings were independently confirmed in three different isogenic clones using QPM on live, attached low-density cultures (*Figure 1—figure supplement 5D*). Based on our measurements there is a ~ 20% increase in dry cell mass in cells expressing YAP5SA. Thus, it is fair to say that the expression of YAP5SA increases cell size in all phases of the cell cycle in a highly reproducible manner independent of the choice of clone, staining errors or detachment of cells from their substrate. However, the exact magnitude of change in size varied somewhat depending on the measurement method of choice. Further experiments using non-isogenic HEK293 cells expressing GFP-tagged YAPWT vs. nGFP demonstrated that YAPWT overexpression only increased cell size in low density cultures but not in high density cultures, a condition where YAP is excluded from the nucleus and thus transcriptionally inactive

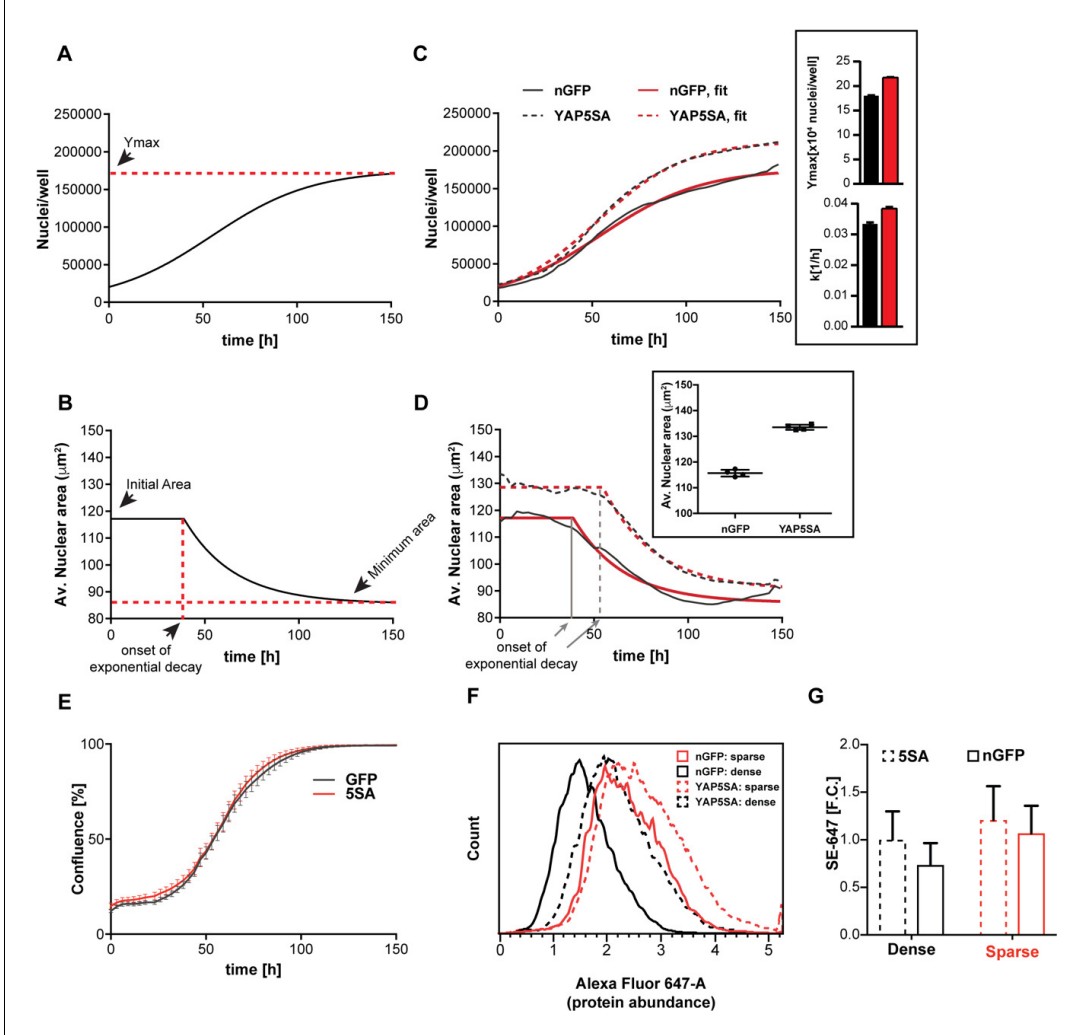

**Figure 1.** YAP5SA expression affects cell size and population growth dynamics. (A) Example of population growth dynamics. The figure is a fit of data to the logistic growth equation in *Figure 1—figure supplement 3A*. Ymax is the carrying capacity of the population. (B) An example of changes in average nuclear area in a population of unsynchronized Flipin-Trex-293 cells over time after fitting to a plateau followed by exponential decay. While nuclear area is initially constant, it exponentially decays to a minimum as cell number increases. (C) Population growth curves of FlipinTrex-293 cells expressing constitutively active YAP (YAP5SA) or nuclear-GFP (nGFP). FlipinTrex-293 cells were seeded at ~20,000 cells/well on a 96-well plate (low density) and monitored over time. Ymax and k are both higher in 5SA cultures than nGFP cultures. In black, the average of 4 wells; in red, the fit to a logistic growth model; n = 4; mean ± SEM. (D) Changes in nuclear area over time in the same populations of cells in (C). Nuclear area is larger in YAP5SA cells vs. controls, but still decreases exponentially as cell density increases. In black, the average of 4 replicates; in red, the fit to a plateau followed by an exponential decay; n = 4; mean ± SEM. (E) Cell confluence is estimated by the relative area of the culture vessel covered by cells in bright field images. We observe no significant difference in the confluence of the YAP5SA vs nGFP cells. (F) Protein content was compared between populations of YAP5SA and nGFP ~20 hr after the same number of cells were seeded in 10 ml medium on 10 cm-dishes to represent either low density (~25% confluence) or high density (>90% confluence) conditions. All samples were simultaneously trypsinized, permeabilized, stained and measured on a LSRII flow cytometer at a concentration of 1 × 10⁶ cells/ml. A total of 10,000 cells were analyzed per condition. Protein content is higher in YAP5SA cells whether sparsely or densely seeded. (G) The population mean and SD of the data in (F).

The online version of this article includes the following figure supplement(s) for figure 1:

**Figure supplement 1.** Example of images acquired on and analyzed by the Incucyte ZOOM to obtain data about the number of nuclei and their average area.

**Figure supplement 2.** Nuclear area is a good proxy for cell size in Flipin-Trex-293 cells.

**Figure supplement 3.** Fitting changes in cell number over time using the Logistic growth equation.

**Figure supplement 4.** Culture density affects cell death rates and cell size.

**Figure supplement 5.** The effect of YAP5SA, YAPwt and YAPsh expression on cell size.

**Figure supplement 6.** Serum is required for the 5SA-dependent changes in proliferation and size.

(*Figure 1—figure supplement 5E*). We also confirmed that knocking down YAP by 80% in WT cells by constitutively expressing YAP-targeting shRNA (YAPKD cells; *Figure 1—figure supplement 5F*) decreased cell size (*Figure 1—figure supplement 5G*) as previously reported (*Hansen et al., 2015*). Through the remainder of the manuscript, we mainly focus on the most dramatic changes in size caused by YAP5SA in sub-confluent cultures.

## The non-cell autonomous nature of YAP-mediated changes in cell size

In isogenic Flipin-Trex cells, YAP5SA or nGFP expression is doxycycline (Dox)-inducible. Therefore, we would not expect that in the absence of Dox there would be any difference in cell size between uninduced YAP5SA, nGFP and parental cells. To our surprise in the absence of Dox the average cell mass in the three YAP5SA-expressing clones was significantly higher than in nGFP clones and the parental cells (9% vs. 20% when cells are cultured in Dox for 4 days; *Figure 2A* vs. *Figure 1—figure supplement 5D*). It is well known that Dox-inducible expression systems can be leaky, so one possible explanation for the larger size would be that the <1 % cells expressing YAP5SA in the absence of Dox could be affecting the behavior of the population as a whole. To test this possibility, we compared the size of parental FlipinTrex-293 cells (WT) when co-cultured together with cells expressing GFP-tagged YAP5SA or nGFP (*Figure 2B*). We found that when cultured in the same well WT cells were similar in size to the YAP5SA cells, both of which were ~30% larger than the nGFP cells and their co-cultured WTs (*Figure 2C*). Thus, although the expression of YAP5SA is obviously cell autonomous, its effect on size is shared with untransfected cells in the same well.

Non-cell autonomous effects could be mediated either by cell-cell contact, or by diffusible molecules (secreted into or depleted from culture medium). To investigate whether these effects required cell-cell contact, WT Flipin-Trex-293 cells were seeded on the bottom of a well, physically separated from a population of feeder cells expressing YAP5SA or nGFP in a Transwell chamber (*Figure 2D*). In this condition both cell types share and exchange components in the medium but do not physically contact each other. When cell size was measured by protein content, cells co-cultured with YAP5SA cells were ~50% larger compared with those co-cultured with nGFP controls (*Figure 2E*). This suggests that YAP's non-cell autonomous effect on cell size is likely mediated by a soluble factor(s) produced by YAP5SA-expressing cells. Based on these results we would expect that conditioned medium from 5SA-expressing cells, when added to naïve cells, would result in a size increase, which it did (*Figure 2F*). Conditioned medium also increased the size of NIH 3T3 fibroblasts after a 48 hr treatment (*Figure 2—figure supplement 1A,B*).

## The effect of YAP5SA on cell number is mediated both by cell autonomous and non-autonomous processes

To investigate whether YAP expression affects cell number as well as cell size in a non-autonomous manner, we co-cultured cells expressing n-mCherry 'recipients' with cells expressing either YAP5SA or nGFP only 'feeders' (*Figure 3A*). The two cell types were well-mixed keeping the total cell count constant while varying the ratio between the recipient and feeder cells. We found that recipient cells grew to a higher number ($Ymax$) when YAP5SA cells were in large excess (four to eight times as many), compared to being co-cultured with nGFP controls (*Figure 3B,C*). The population growth rate at low density ($k$) was not affected by the number of YAP5SA expressing cells (*Figure 3D*).

We suspected the difference between $k$ and $Ymax$ in the presence of YAP5SA was because its effect on $k$ is cell autonomous, while its effect on $Ymax$ is not. If this hypothesis were true, k would be lower in YAPKD cells irrespective of what cells they were co-cultured with. In contrast $Ymax$ should increase in YAPKDs when co-cultured with YAP5SA feeders. Indeed, k was lower in YAPKDs vs controls, and co-culturing either cell type with YAP5SA cells had little effect on $k$, while significantly increasing $Ymax$ (*Figure 3E–G*). These results are most easily explained if the effect of YAP on $k$ was indeed cell autonomous, whereas its effect on $Ymax$ was non-cell autonomous. Furthermore, we show that treating serum-starved FlipIn-Trex-293 or 3T3 cells with 5SA-conditioned medium increased the number of cells in dense cultures vs. the addition of conditioned medium derived from nGFP cells (*Figure 3H* and *Figure 1—figure supplement 5D,E*). These results also support a YAP-dependent, non-cell autonomous mode of regulating $Ymax$.

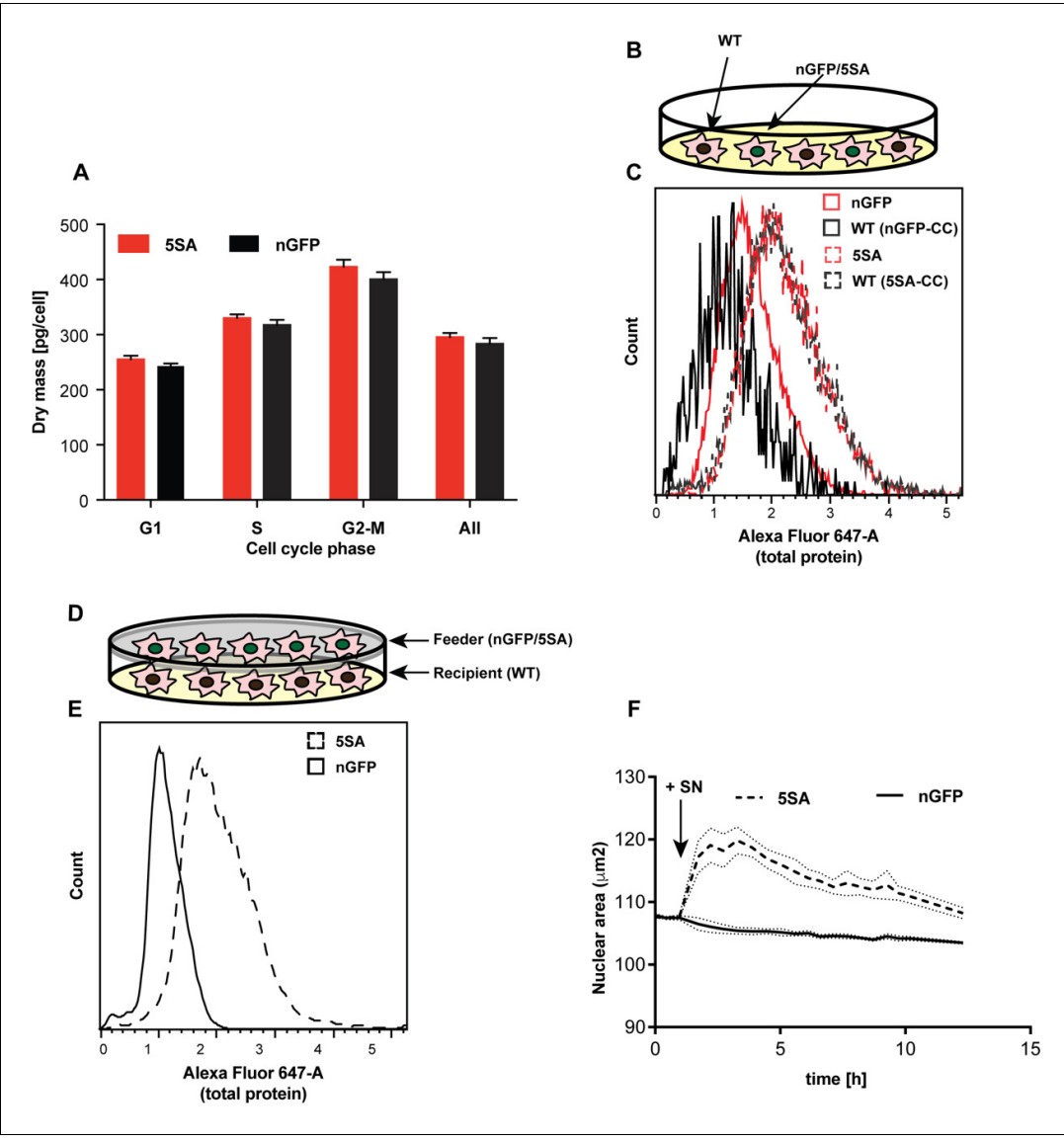

**Figure 2.** YAP5SA increases cell size non-autonomously. (**A**) The average dry mass/cell ± S.D. in cultures of 3 isogenic clones of Flipin-Trex-293 cells engineered to express YAP5SA or nGFP was measured using quantitative phase microscopy (QPM). Cells had never been cultured in the presence of doxycycline before they were transferred to a Fibronectin-coated, glass-bottom 6-well plate at ~50,000 cells/well for the measurement. 30 min before the measurement Hoechst was added at a concentration of 2 µg/ml to allow DNA staining of live cells. The overall average per clone was calculated from >4,000 cells. The average size per cell cycle phase was determined from cells binned into the respective cell cycle phase depending on Hoechst-levels. Cell size is slightly higher in the population of uninduced 5SA cells vs. controls. (**B**) Experimental setup of data in (**C**). In short, cells expressing nuclear GFP (nGFP) or GFP-tagged YAP5SA (YAP phospho-mutant) cells were intermixed with wildtype (WT) cells at a ratio of 2:1 and grown for ~24 hr on 10 cm dishes before they were trypsinized and processed as single-cell suspensions at a concentration of $1 \times 106$ cells/ml. Total cell number was kept the same between both cultures, and all samples were simultaneously prepared for labelling using the SE-647 protein dye and analyzed on a LSRII. (**C**) Histograms of average protein content as estimated by SE-657 signal per cell. Cells were separated into two groups depending on whether or not they expressed GFP. Cc denotes the GFP-negative WT cells in the culture. GFP+ (red dotted line) and GFP- cells (black dotted line) in YAP5SA-co-cultures are larger than cells in the nGFP co-cultures. (**D**) Experimental set up for data in (**E**). WT cells were co-cultured with YAP5SA or nGFP cells on a 0.4 µm Transwell membrane in a 10 cm dish. The membrane allows the exchange of medium components but not physical contact. (**E**) WT cells from the setup in (**D**) were processed as explained above to allow SE-dye based quantification of their protein content. WT cells co-cultured with YAP5SA cells have higher protein content than those co-cultured with nGFP cells. (**F**) FlipinTrex-293 cells expressing n-mCherry were starved for 6 days in 0.5%

*Figure 2 continued on next page*

*Figure 2 continued*
FBS before they were treated with equal amounts of 5SA- or nGFP-conditioned medium. Medium conditioned by 5SA cells but not nGFP cells increases nuclear area.
The online version of this article includes the following figure supplement(s) for figure 2:

**Figure supplement 1.** The effect of 5SA-conditioned medium on 3T3 fibroblasts.

## Many proteins affected by YAP5SA overexpression are secreted or membrane proteins

We have tested the notion that these non-cell autonomous effects are mediated by secreted proteins that are transcriptional targets of YAP. Using RNA-Seq and mass spectrometry we compared the transcriptional and corresponding protein expression changes in FlipinTrex-293 cells in response to YAP5SA expression (*Figure 4A*, *Supplementary file 1*). We concluded that extracellular proteins were a large portion of those expressed in response to YAP5SA (eight fold enrichment relative to all detected proteins, p-value$<10^{-12}$; *Figure 4B*). While in some cases the change in RNA and protein levels were similar, in many other cases they were not. For example, CTGF, CYR61 and AMOTL2, which are canonical YAP targets, were elevated at the protein and mRNA levels, while we see an almost four-fold decrease in LOX and TNC proteins, with no change on the mRNA level (*Figure 4A*). CYR61 and CTGF are members of the CCN family of matricellular proteins, non-structural proteins that reside in the extracellular matrix and are thought to regulate cellular processes like migration, apoptosis, division and differentiation, involving the coordination of several signaling pathways (*Holbourn et al., 2008*). Though both genes have previously been used as robust transcriptional readouts for YAP activity (*Dupont et al., 2011*; *Yimlamai et al., 2014*), their contribution to the YAP-dependent changes in cell behavior has not been clarified. Specifically, their role in coordinating YAP-dependent cell proliferation, apoptosis and size control has not been explicitly studied. We also considered whether Amphiregulin, a secreted molecule related to EGF, that had previously been identified as a potential mediator of YAP-dependent non-cell autonomous growth in 3D cultures of MCF10 cells (*Zhang et al., 2009*) was involved in YAP-dependent size or proliferation control in our cells. In these experiments, Amphiregulin RNA was expressed at a very low level (FPKM <1 compared with FPKM >100 for CYR61); and was not identified by mass spectrometry. By contrast we identified 3 CYR61 and 6 CTGF peptides.

## The YAP target, CYR61, affects cell number but not cell size

To test whether CYR61 and/or CTGF mediated any of the non-cell autonomous changes in cell size or number, we used neutralizing antibodies to inhibit their extracellular activity in co-cultures of n-mcherry recipient cells and YAP5SA cells. The identity of the recombinant proteins and the specificity of the antibodies we used were demonstrated by their mutual recognition in *Figure 5—figure supplement 1*. Neutralizing CTGF had a small effect on cell number, mildly decreasing $k$ (~12%) while mildly increasing $Ymax$ (~12%) of the n-mCherry cells in the co-cultures (*Figure 5A–C*). On the other hand, neutralizing CYR61 significantly decreased $Ymax$ (up to ~50%) but increased $k$ (up to ~200%) in a dose-dependent manner (*Figure 5D–F*). Our RNA-Seq data suggest that the amount of CYR61 (FPKM >100) transcribed in these cultures is greater than CTGF (FPKM <50). Consistent with the difference in expression levels, increasing the concentration of antibody had a much larger effects for anti-CYR61, as compared to CTGF (*Figure 5A,D*). These findings support a major role for CYR61 in the non-cell autonomous, YAP-mediated increase in $Ymax$. However, the increase in the rate of population growth observed at low cell density appears not to be mediated by CYR61, consistent with the measurements in *Figure 3* that show that the dominant effects on $k$ are cell autonomous.

To test whether YAP expression in the responding cells is required for the CYR61-dependent increase in Ymax, we knocked down YAP and added CYR61. As Shown in *Figure 5G–L* YAPKDs as well as controls responded by increasing $Ymax$ by ~30% and 20% respectively, but decreased k by about 10% and 50% (*Figure 5G–L*), confirming that CYR61's non-cell autonomous influence on cell number – specifically the carrying capacity ($Ymax$) – does not require active YAP in the responding cells. These results were therefore consistent with the co-culture experiments (*Figure 3E–G*). Finally, neither CTGF or CYR61 neutralization nor addition affected nuclear area (*Figure 5M–P*); hence

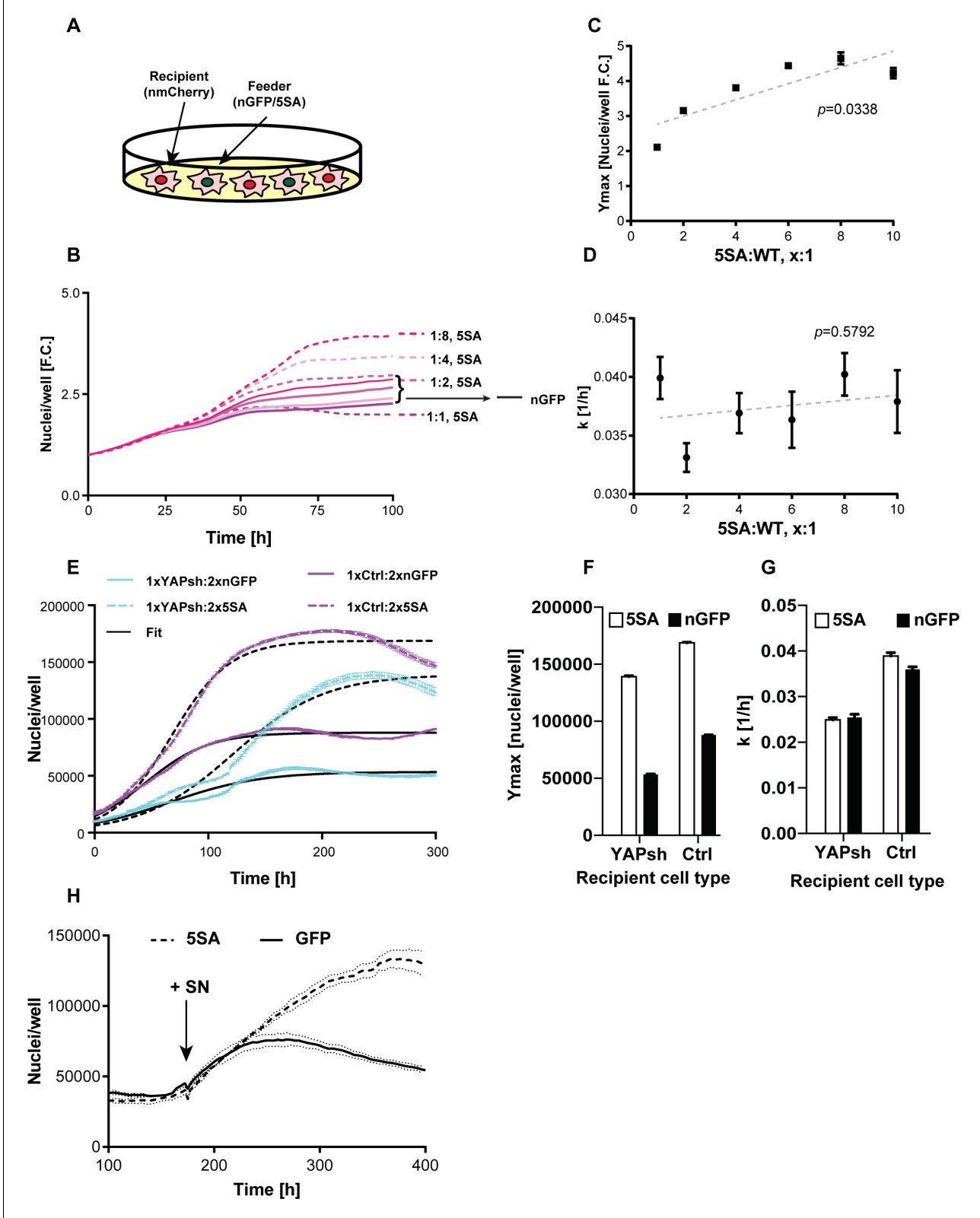

**Figure 3.** YAP affects the population growth rate (k) and carrying capacity (Ymax). (**A**) Experimental design for results in (**B–G**). Flipin-Trex-293 cells expressing nuclear GFP (nGFP) or GFP-tagged YAP5SA (YAP phospho-mutant) work as feeder cells for intermixed recipients expressing nuclear mCherry (n-mCherry). The total number of cells per well at seeding is kept constant at ~20,000 cells/well on a 96-well plate, though the ratios between cell types varied. Ymax and k are estimated based on counts from the recipient population only. (**B–D**) Increasing the fraction of YAP5SA to n-mCherry

*Figure 3 continued on next page*

*Figure 3 continued*

cells increases Ymax but does not affect k; *p-values* indicate the likelihood that the slope is non-zero; n = 5 wells; mean ± SEM. (**B**) Depicts raw data. (**E**) Population growth curves of YAP knockdowns (light blue) or empty vector controls (Ctrl; purple) when co-cultured with twice the number YAP5SA (dashed lines) or nGFP (solid line) cells; n = 6; mean ± SEM. Black lines represent the fit of the data to a logistic model. (**F**) Ymax is higher when recipient cells are co-cultured with YAP5SA feeders even if recipients have reduced levels of endogenous YAP (YAPKDs). Ymax is slightly lower than controls in YAPKDs. (**G**) K is lower in YAPKD recipients than ctrls, and is unaffected by feeder cell type. Meanwhile, co-culturing ctrls with YAP5SA feeders as depicted in (**A**) does increase k mildly. (**H**) Cells expressing n-mCherry were starved in 0.5% FBS for 6 days before they were treated with 5SA- or nGFP-conditioned medium. Treating cells with YAP5SA-conditioned medium increases Ymax in comparison to nGFP-conditioned medium; n = 4; mean ± SEM.

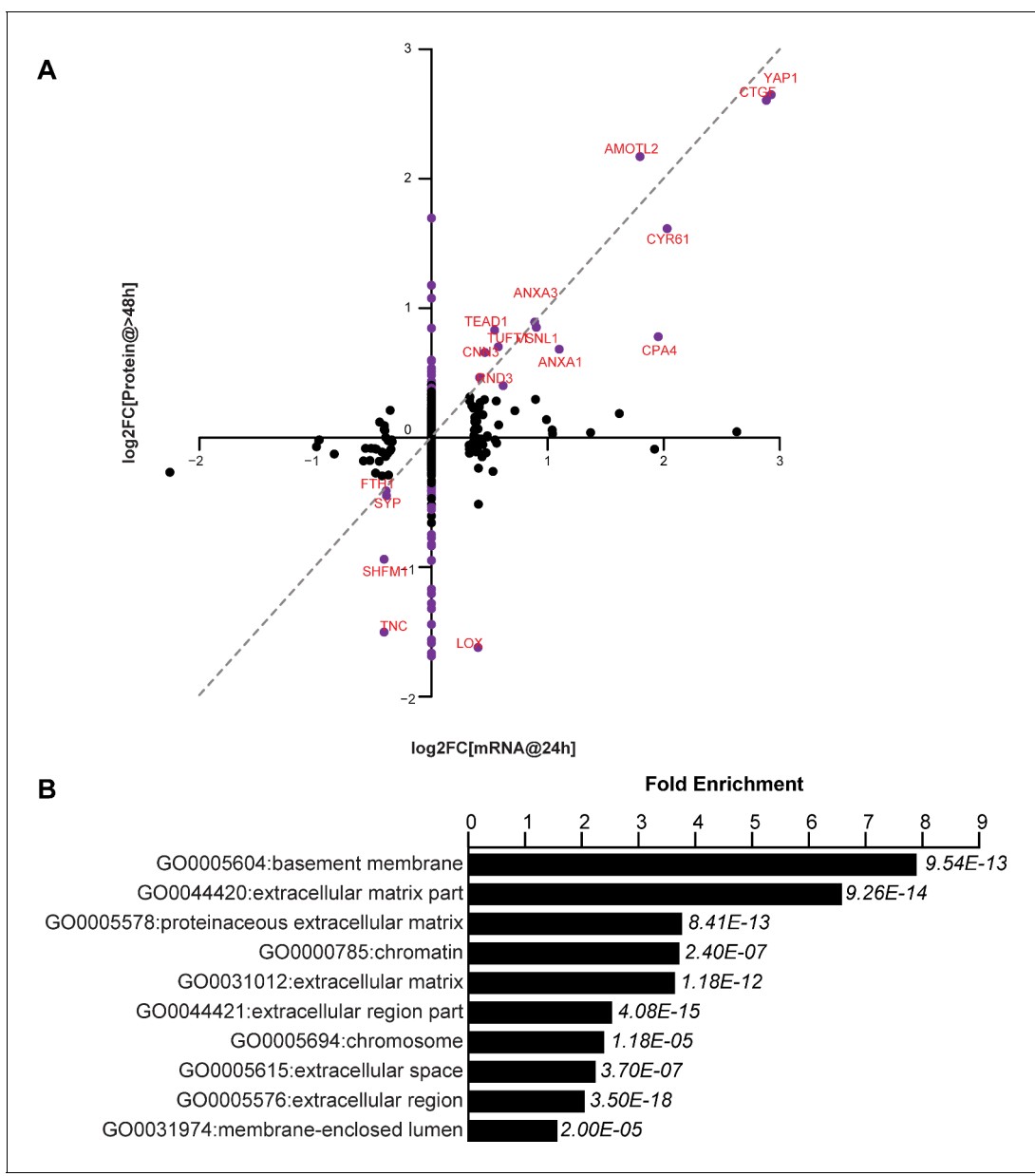

**Figure 4.** YAP-dependent changes in protein and mRNA expression levels. (**A**) Scatter plot of changes in protein (y-axis) vs. mRNA levels (x-axis) upon YAP expression. Dots in purple indicate proteins have a log2[fold change] (log2FC) more than two standard deviations away from the mean. Dotted line indicates a 1:1 change in protein vs. mRNA levels. (**B**) The 10 most significantly enriched cellular compartments in which regulated proteins reside (according to fold enrichment relative to all identified proteins). Benjamini-Hochberg *adjusted p-values* are indicated in italics.

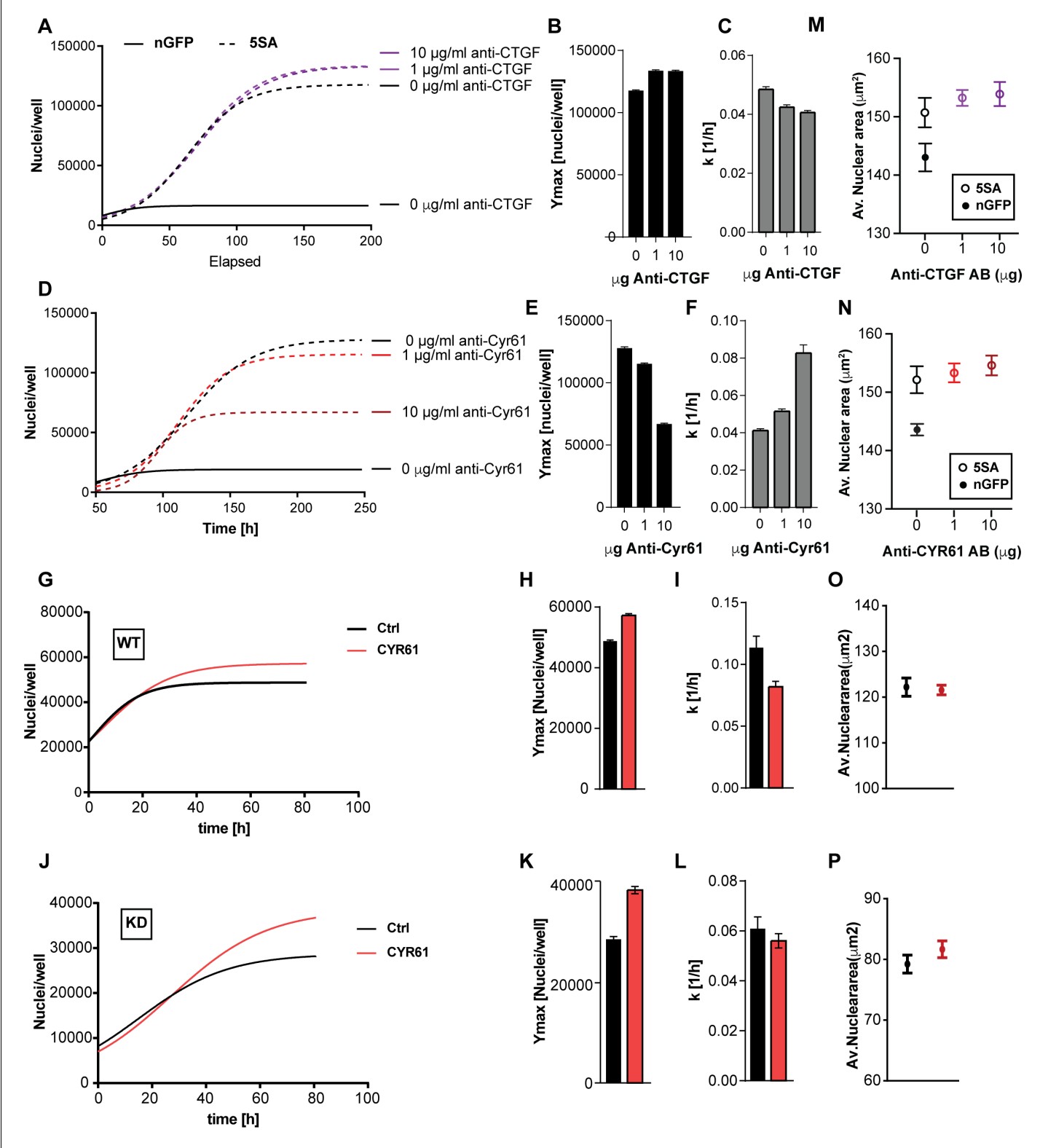

**Figure 5.** The effects of CTGF and CYR61 on cell size and number. CTGF or CYR61 were neutralized by the addition of antibodies at the indicated concentration to the culture medium when cells were seeded. (**A–C**) In co-cultures of 6x YAP5SA: 1x nuclear mCherry expressing cells (n-mCherry), CTGF neutralization increases the n-mCherry cells' carrying capacity (Ymax), while decreasing their rate of growth (k). (**D–F**) Neutralizing CYR61 in these cultures has opposing effects of larger magnitude compared to CTGF. (**G–I**) Treating cells expressing empty vector (WT) with 1 μg/ml of CYR61 increases Ymax and decreases k. (**J–L**) YAPKD cells respond to exogenous CYR61 with an increase in Ymax and a decrease in k. (**M–N**) Cell size is not

*Figure 5 continued on next page*

*Figure 5 continued*

affected by the presence of anti-Cyr61 or anti-CTGF. (**O, P**) Adding recombinant CYR61 at 1 µg/ml also has little effect on cell size in YAPKDs and controls. (**B**), (**C**), (**E**), (**F**), (**H**), (**I**), (**K**), (**L**) n = 5 wells; mean ± SEM. (**M–P**) n = 5 wells; mean ± SD).

The online version of this article includes the following figure supplement(s) for figure 5:

**Figure supplement 1.** Validation of anti-CYR61 and anti-CTGF antibodies.

some factors other than CTCF or CYR61 must be responsible for the non-autonomous effects of YAP on cell size.

## Cyr61 increases Ymax by reducing apoptosis

Ymax is clearly different between nGFP and 5SA cells (*Figure 1C*), although the distribution of cells in the cell cycle suggests no difference in proliferation rates between both cell types at high density (*Figure 6A*). We hypothesized the striking effect of CYR61 on *Ymax* is most easily explained by an effect on cell death, which occurs at higher rates in high vs. low density cultures (*Figure 1—figure supplement 1–4A,B*). Consistent with that, in the presence of anti-Cyr61 antibody, there is a three-fold increase in cell death in cultures of Flipin-Trex-293 cells, which can be reduced to control levels upon addition of an anti-apoptotic peptide (Q-VD-OPH; *Figure 6B*). We also observed that cells treated with anti-CYR61 were mostly all dead by the end of 6 days (*Figure 6C*). Cells treated with CYR61 or IgG were almost 100% alive with many still cycling. We conclude that high levels of CYR61 reduce density-dependent apoptotic cell death in YAP5SA cells.

## Discussion

In the early studies of cell growth, the focus was on unicellular organisms or cells grown at low density. Studying cell growth at high density meant having to deal with complex interactions of the individual components, the space they occupy, and effects on the solvent or the medium, all of which are less confounding at low density. Nevertheless, interest in growth at high cell density was piqued when biologists made 2D cultures of animal cells on petri dishes. For example, chicken embryo fibroblasts grown in such a manner generate a monolayer and stop growing (*Weinberg, 2007*). It is unlikely that this was simply depletion of the medium, for when Rous Sarcoma Virus was added the quiescent cells were transformed and proliferated, overgrowing the monolayer and producing three dimensional 'foci' on top of the monolayer (*Groupe and Manaker, 1956*). There are two phenomena here to consider, the still unexplained tendency of cells to stop growing upon crowding and the tendency of a few infected or transformed cells to overcome whatever inhibitions were observed in the uninfected monolayers. With the passage of 50 years these phenomena have neither been completely resolved nor completely forgotten. The problem of cell interactions is still a serious topic in cancer (*Kamińska et al., 2015*) and more recently has emerged in many other biological settings, such as the behavior of cells in regeneration or in stem cell niches (*Lane et al., 2014*).

More interest in the reciprocal interactions of proliferating and resident cell populations was stirred by the discovery of the Hippo pathway, originally discovered in *Drosophila*, and later shown to be activated in high-density cultures. The major effector of the pathway is the transcription regulator YAP, which is formally considered an oncogene, because it stimulates growth (*Dong et al., 2007*). Over-activating YAP results in a breakdown of some of the usual barriers to growth in complex tissues and in high density cell cultures (*Zhao et al., 2007*). An interesting feature of the pathway is its sensitivity to alterations in the nature of the substratum, such as ECM composition and stiffness (*Dupont et al., 2011*; *Kim and Gumbiner, 2015*). It is still generally assumed that the aggressive behavior of the cells is governed in a cell autonomous manner – like the original Rous transformation mentioned above. Another, and more unique feature of YAP is its control of organ size. Organ size is determined by a tight coordination of cell death, proliferation and sometimes differentiation as well as cell size. Organ size control is best studied in vivo where the complexity of cell-cell and cell-matrix interactions are better maintained, but this complexity makes teasing apart underlying mechanisms challenging. Cultured cells can be used to model some of the aspects of organ size in the whole animal but lacks the more complex context. Nevertheless, as cultured

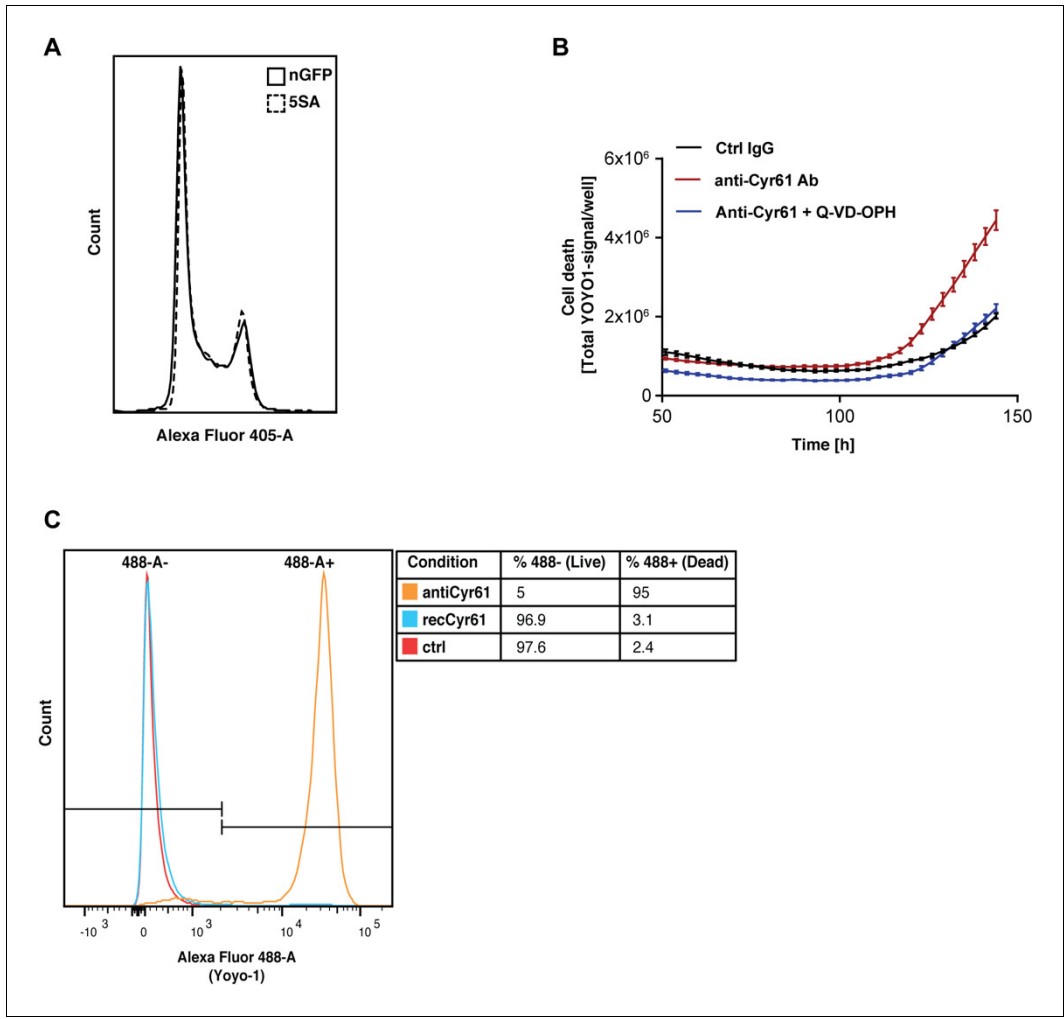

**Figure 6.** CYR61 affects cell proliferation and death at high density. (**A**) FlipinTrex-239 cells expressing YAP5SA (5SA) or nuclear GFP (nGFP) were seeded at a density of $10^7$ cells per 10 cm dish which is equivalent to the number of cells on a fully confluent dish. ~24 hr later, cells were trypsinized, treated with the cell-permeable DNA dye Hoechst 3342, and analyzed on a LSRII. The distribution of FlipinTrex-293 cells in the different cell cycle phases is comparable between these cultures. (**B**) Cells were seeded in the presence of 50 nM of the green fluorescent, non-cell permeable DNA dye YOYO-1; 10 µg/ml of anti-CYR61 or polyclonal rabbit IgG as a conrol; and 10 µM of the anti-apoptotic peptide Q-VD-OPH. Cell death increases as cell density increases as indicated by the increase in green fluorescent signal per well. Cell death is exacerbated in the presence of anti-Cyr61, and can be restored to control levels in the presence of the anti-apoptotic peptide Q-VD-OPH. (**C**) FlipinTrex-293 cells were cultured at high density cultures in the presence of YOYO-1 and treated with 10 µg/ml of anti-Cyr61/IgG or 1 µg/ml of recombinant CYR61 for 6 days. By that time, most of the cells treated with anti-CYR61 are mostly dead, while IgG controls and those treated with recombinant CYR61 are mostly alive.

systems become more complex, there is an expectation that mechanistic features of organ and tissue growth can be recognized and understood.

In this paper we have extended observations to cell autonomous and non-cell autonomous signaling downstream of YAP in cultured cells. In this system we have examined the growth of cells at different cell densities; the effects of cell proliferation and cell death; and the combined effects of proliferation and cell size. By such measurements we were able to gain a clearer understanding of YAP's role in regulating these phenomena on a cellular as well as population level. We found that the growth rate of populations, *k*, at low density, was elevated significantly when cells expressed a constitutively active form of YAP, YAP5SA. compared to wild type (*Figure 1B*) cultures. The effect

on k was cell autonomous and driven predominantly by a change in proliferation rates. This increase in proliferation was accompanied by an increase in the fraction of cells in S-G2-M phases of the cell cycle and may be explained by previous reports that there is a YAP-mediated increase in the cell cycle regulators CyclinE and Cyclin D, and consequently a higher rate of entry into S-phase (*Dong et al., 2007*; *Shu and Deng, 2017*; *Tapon et al., 2002*).

Even more obvious was YAP5SA's effect on the number of cells at high density, where the carrying capacity, *Ymax*, increased about 35%. The effect of YAP on Ymax was demonstrably non-cell autonomous and mediated by CYR61, which increases Ymax even in the absence of endogenous YAP indicating that it is not simply an autocrine mediator. CYR61 is one of the strongest genes we find differentially expressed on both the protein and mRNA level. It is a common YAP target that is a member of the CCN group of matricellular proteins, known to be involved in tissue remodeling (*Chen et al., 2001*), inflammation and embryogenesis (*Katsube et al., 2009*; *Latinkic et al., 2003*). In our experiments, the effects of CYR61 on cell number at high density are largely a result of a reduction in apoptosis, while it seems to suppress proliferation at low density. Previously, CYR61 was shown to reduce apoptosis in MCF7 cells (*Lin et al., 2004*) and is one of the earliest induced genes in response to a partial hepatectomy, which also involves/requires YAP (*Grijalva et al., 2014*; *Lu et al., 2018*; *Su et al., 2002*). Only recently has CYR61 activity in hepatocytes been implicated in YAP-mediated liver fibrosis by activating macrophages (*Mooring et al., 2020*), suggesting that CYR61 is more than a YAP reporter in that context, and is of importance in mediating non-cell autonomous YAP activity in vivo. CYR61 was also reported to play a role in inhibiting hepatocyte proliferation in response to carcinogen-mediated injury (*Chen et al., 2016*). Altogether, this suggests that CYR61 plays various roles in cell culture as well as in vivo, which could depend on the cell type as well as the particular context in which CYR61 is activate. This is particularly interesting in the whole-organ context, where the coordination of interactions between cell types is often necessary and would be expected to depend on cell non-autonomous signaling. The present study involved only a single type of cell examined in two contexts as responders and signalers. Exploring the role of YAP and its extracellular targets in mixed cultures of different cell types could provide additional insight into its role in complex tissues.

Finally, cell size, also seems to be regulated non-cell autonomously upon YAP expression. We applied several modes of measurement: nuclear area, volume, protein content and dry mass, all of which are often used as metrics to describe cell size, and they all show that expressing YAP5SA increases cell size between ~17–30%. We believe the changes in nuclear area and dry mass (17% vs. 20% respectively) are the most reliable and are also well-correlated. We tested whether the two extracellular proteins most affected by YAP expression, CYR61 and CTGF, have an effect on WT cell size. Neither the depletion of either protein nor the addition of the recombinant proteins affected cells size. We did not find evidence that the classic size regulating growth factors IGF1/2 are produced to any appreciable degree by our cells. There are, however, many candidate secreted proteins that are stimulated by YAP and which should be tested in the future.

The YAP targets, CTGF and CYR61, both members of the CCN family of matricellular proteins, have opposing effects on cell number in co-cultures with YAP5SA expressing cells. The levels of CTGF are small compared with CYR61 (*Figure 5*) which we believe explains why the observed non-cell autonomous effects on population growth are mainly those mediated by CYR61. CCN proteins, unlike typical growth factors, are not known to bind to any receptor tyrosine kinases, but rather to integrins and other growth factors, thereby mediating the activity of those conventional growth factors (*Leask and Abraham, 2006*). With low serum in the medium (<1%), Ymax decreases (*Figure 1—figure supplement 3E*). Size is also affected by the level of serum (*Figure 1—figure supplement 6B*), suggesting that another matricellular factor might mediate YAP-dependent changes in cell size.

Work by Tian and colleagues (*Tian et al., 2013*) had previously demonstrated that a high molecular weight hyaluronic acid produced by naked mole rate fibroblasts contributes to early contact inhibition; the equivalent of a decrease in the population's carrying capacity. This form of hyaluronic acid acts via the upstream Hippo component NF2, which inhibits YAP activity. Combined with our results on secreted factors, such findings suggest that extracellular cues are an important part of the regulation of both cell and organ (tissue) size. The non-cell autonomous nature of YAP signaling would allow local signals to affect remote cells of the same or different type as is expected to occur in regeneration. In other words, the activation of YAP in a small fraction of cells at the site of injury could through secreted signals engage a wider portion of the tissue or even a whole organ to

participate in tissue repair. The receiving cells might also secrete signals as part of their response, further reinforcing the signals and allowing them to propagate farther than would be possible by diffusion alone.

## Materials and methods

### Cloning

pEGFP-C3-YAP5SA plasmids were generated from the pEGFP-C3-hYAP1 plasmid gifted from Marius Sudol (Addgene plasmid # 17843; RRID:Addgene_17843 *Basu et al., 2003*). YAP amino acids S61, S109, S127, S164, S381 were all mutated to Alanine using the QuikChange II XL Site-Directed Mutagenesis Kit (Agilent). The following nuclear localization signal atggatccgaagaaaaaacgtaaaggccgtatg-gatccgaagaaaaaacgtaaagg-ccgt was appended to the 3'-end of GFP in the pAcGFP1-C3 vector (Clontech) to ensure its nuclear localization.

### Cell line generation

All cell lines were maintained in DMEM (Gibco, 10569010) supplemented with 10% FBS (Gibco), 1% P/S (Gibco) unless otherwise stated. Dox-inducible HEK293 cell lines of nGFP or GFP-tagged YAP5SA were generated by sub-cloning either construct into the pcDNA5/FRT expression vector (Invitrogen) then co-transfecting the generated plasmids with the pOG44 Flp-Recombinase Expression Vector to ensure recombination into the genome of Flipin-TRex −293 cells (Invitrogen). Selection with Hygromycin and single clone isolation was followed by a test for Zeocin-sensitivity to ensure proper integration into the genome. Cells were maintained in Doxycycline unless otherwise specified.

YAPsh and control constructs were generously provided by Taran Gujral (*Gujral and Kirschner, 2017*). After transfection of HEK293 cells (ATCC Cat# CRL-1573, RRID:CVCL_0045) using PEI-40 and antibiotic selection in Puromycin for two weeks, isolated single colonies were observed and were individually isolated. To label cells with nuclear localized mCherry we used Lentiviral infection. First nuclear-mCherry was excised from pBRY-nuclear mCherry-IRES-PURO (Addgene plasmid #52409, RRID:Addgene_52409) by digestion with EcoRI and NotI and ligated into lentiviral expression plasmid pLVX-Ef1α-N1-mCherry. Lentiviruses were then produced via Fugene six co-transfection of psPAX2 (Addgene plasmid #12260, RRID:Addgene_12260) and pMD2.G (Addgene plasmid #12259, RRID:Addgene_12259) into 293 T cells ((ATCC Cat# CRL-11268, RRID:CVCL_1926). The harvested lentivirus was then used to infect the desired cell line. One week after cells were infected with Lentivirus, they were FACS sorted for mCherry expression.

### Transwell assays

Transwell assays were done using 75 mm Transwell plates with a 0.4 μm Pore Polycarbonate Membrane Insert (Corning). Recipient cells were cultured on the bottom of the plates while feeder cells were cultured on the insert for 72 hr.

### Antibodies and recombinant proteins

Neutralization antibodies against CYR61 (mainly NOVUS biological NB100-356, RRID:AB_10000986 raised against a peptide of amino acids 150–250 of human CYR61, validated independently with Abnova PAB12526, RRID:AB_10550916 raised against a peptide of amino acids 250–350 of human CYR61) or CTGF (Peprotech 500-P252, RRID:AB_1268211) were used at the indicated concentrations in the respective figures. A polyclonal Rabbit IgG Antibody was used as a control (Protein tech; 30000–0-AP, RRID:AB_2819035). Recombinant proteins CTGF (Peprotech), CYR61 (Peprotech) and Insulin (Peprotech) were used at the concentrations and under the conditions indicated in the respective figure legends.

### Conditioned medium production

FlipinTrex-cells were cultured in Nunc TripleFlasks with DMEM plus 10% FBS and 1% P/S until they reached 100% confluence. Cells were then washed in PBS and kept in medium with 0.5% FBS and Dox for 6 days. Medium was collected, spun down and the supernatant then filtered on a 0.2 μm membrane to remove cell debris. The medium was concentrated using 3 kD cut-off Centricon Plus-

70 filter devices (Millipore). The concentrate was kept at −20°C until further use. For protein concentration estimation, the concentrate was diluted to within working range before a BCA assay (Thermo Scientific) was performed against a standard.

### Incycyte-based image acquisition and automated image analysis for nuclear segmentation

Images from four different regions per well were acquired at regular intervals using a 10x lens (Nikon, MRH00101) from cells cultured on 96-well plates (Eppendorf) in an Incucyte Zoom apparatus installed in a regular tissue culture incubator. After image acquisition, the red signal was used for nuclear segmentation using Incucyte ZOOM's automated image analysis software. Images from all wells and across the entire time course were segmented using the same parameters. The number of red objects was used to represent the total number of nuclei. The average area estimated from the segmented objects in the red channel represent the average nuclear area.

### Quantitative phase microscopy (QPM)

50,000 Flipin-Trex-293 cells were seeded on a Fibronectin-coated (50 ug/ml) glass-bottom 6-well plate (Cellvis) ~18 hr before imaging. This ensured that cells were at a low-enough density to allow high-quality segmentation of ~4000 single cells per well. Cells were cultured with 2 µg/ml Hoechst for 30' before imaging. Imaging and analysis was done as described in *Liu et al. (2019)*. Briefly Cells were maintained at 36°C in 5% CO2. The phase images were taken using a 10x lens of N.A. 0.30 (Nikon, MRH10101) using the SID4BIO camera (Phasics) integrated into a Nikon Eclipse Ti microscope with a halogen lamp as the transmitted light source. A C-HGFI mercury lamp was used for fluorescence illumination. All the image processing was done using a custom MATLAB script executed on a high-performance computer cluster.

### Data plotting and fitting

Data were imported into Prism7 (Graph Pad) where their visualization, fitting and statistical analysis was done.

### Coulter counter measurement

Cell volume was measured on a Coulter Counter (Beckman Coulter). For the measurement cells were trypsinized to single cells then resuspended in 10 ml Isoton diluent buffer (Beckman Coulter).

### Total protein/DNA staining

Cells were trypsinized to a single-cell suspension. $10^6$ cells were counted per condition, thoroughly washed in PBS and fixed for 20 min in ice cold, 4% PFA solution. Cells were then washed in PBS and permeabilized using 0.5% Triton-X for 10 min at room temperature. After repeated washes, cells were kept on ice with a 10 µg/ml Hoechst 3342 and/or 0.4 µg/ml of Alexa Fluor 647 Succinimidyl-Ester (Invitrogen) in a light protected tube for 45 min before they were analyzed on an LSRII flow cytometer (BD Biosciences).

### RNA-Sequencing and data analysis

Flipin-Trex-293 cells expressing Tet-inducible YAP5SA or nGFP were grown to confluence in medium supplemented with Tet-approved FBS (Takara) supplemented with 0 or 5 ng/ml of Dox. Total RNA was harvested and isolated using Qiagen's RNeasy mini kit. Samples were further processed at Harvard's Biopolymer facility as follows: After polyA enrichment of mRNA using Takara's PrepX PolyA mRNA Isolation Kit, libraries were prepared using Illumina's PrepX RNA-Seq Library reagent kit on the Apollo 324 system. Samples were pooled then single-end sequenced on the Illumina NextSeq 500 platform. Sequencing alignment was done using STAR (*Dobin et al., 2013*) and transcriptome assembly and differential expression was done using Cufflinks (*Trapnell et al., 2012*) on a computer cluster. Each comparison group contained at least two independent biological repeats. Data is deposited in GEO under accession number: GSE123296.

## Mass spectrometry and data analysis

Cells were kept under similar conditions as those used for RNA-sequencing. Cellular protein lysate was collected and snap frozen in urea-free lysis buffer supplemented with protease and phosphatase inhibitors (Roche). The supernatant was collected and filtered on a low protein-binding PVDF 0.22 μm membrane to eliminate cell debris, then snap frozen with protease inhibitors (Roche) for further processing. Lysates were reduced with 5 mM DTT, alkylated with 15 mM N-ethylmaleimide for 30 min in the dark. Alkylation reactions were quenched with 50 mM freshly prepared DTT, and proteins precipitated by methanol/chloroform precipitation. Digests were carried out in 200 mM EPPS pH 8.5 in presence of 2% acetonitrile (v/v) with LysC (Wako, 2 mg/ml, used 1:75) for 3 hr at room temperature and the subsequent addition of trypsin (Promega #V5111, stock 1:75) over night at 37°C.

Missed cleavage rate was assayed from a small aliquot by mass spectrometry. For whole proteome analysis, digests were directly labeled with TMT reagents (Thermo Fisher Scientific). Labeling efficiency and TMT ratios were assayed by mass spectrometry, while labeling reactions were stored at −80°C. After quenching of TMT labeling reactions with hydroxylamine, TMT labeling reactions were mixed, solvent evaporated to near completion, and TMT labeled peptides purified and desalted by acidic reversed phase $C_{18}$ chromatography. Peptides were then fractionated by alkaline reversed phase chromatography into 96 fractions and combined into 24 samples. Before mass spectrometric analysis, peptides were desalted over Stage Tips (*Rappsilber et al., 2003*).

Data were collected by a MultiNotch SPS $MS^3$ method (*McAlister et al., 2014*) using an Orbitrap Lumos mass spectrometer (Thermo Fisher Scientific) coupled to a Proxeon EASY-nLC 1000 liquid chromatography (LC) system (Thermo Fisher Scientific). The 100 μm inner diameter capillary column used was packed with $C_{18}$ resin (Accucore 2.6 μm, 150 Å, Thermo Fisher Scientific). Peptides of each fraction were separated over 3–5 hr acidic acetonitrile gradients by LC prior to mass spectrometry (MS) injection. The first sequence scan was an $MS^1$ spectrum (Orbitrap analysis; resolution 120,000; mass range 400–1400 Th). $MS^2$ analysis followed collision-induced dissociation (CID, CE = 35) with a maximum ion injection time of 150 ms and an isolation window of 0.7 Da. To obtain quantitative information, $MS^3$ precursors were fragmented by high-energy collision-induced dissociation (HCD) and analyzed in the Orbitrap at a resolution of 50,000 at 200 Th.

$MS^3$ injection time for phosphopeptides was 150 and 200 ms at a resolution of 50,000. Further details on LC and MS parameters and settings used were described recently (*Paulo et al., 2016a*).

Peptides were searched with a SEQUEST-based in-house software against a human database with a target decoy database strategy and a false discovery rate (FDR) of 1% set for peptide-spectrum matches following filtering by linear discriminant analysis (LDA) and a final collapsed protein-level FDR of 1%. Quantitative information on peptides was derived from $MS^3$ scans. Quant tables were generated requiring an $MS^2$ isolation specificity of >75% for each peptide and a sum of TMT s/n of >0 over all channels for any given peptide and exported as TAB-separated raw text data. The peptide-level sums of TMT s/n signal were integrated into protein-level and converted into relative ratio with 90% confidence intervals using a recently developed method BACIQ - a Bayesian approach to confidence inference for quantitative proteomics. Briefly, this approach reconciles disagreement across multiple peptides and differences in absolute levels of peptide signal, reporting the ratios across conditions for a given protein, modeling quantitative proteomics measurement as a random variable distributed according to hierarchical Dirichlet-Multinomial probability distribution. This method accounts for the absolute level of the peptide signal in a way calibrated for a particular MS instrument. More specifically we used a charge conversion value of 1.7 as previously fitted by us for this instrument and mass resolution (*Peshkin et al., 2019*). Details of the TMT intensity quantification method and further search parameters applied were described in *Paulo et al. (2016b)*.

Gene ontology analysis was done using DAVID (*Huang et al., 2009*) using all identified proteins as background.

## Acknowledgements

We thank Mike Gage for his meticulous assistance with the preparation of the protein samples for mass spectrometry and his overall management of project resources. We thank Jodene Moore of the SysBio FACS facility for her indispensable help setting up FACS analyses on the facility machines. We also thank the BPF Next-Gen Sequencing Core Facility at Harvard Medical School for their expertise and instrument availability that supported this work. We are grateful to the Nikon Imaging

Center at Harvard Medical School for sharing resources; and to the Research Computing Group for support with cluster computing for image processing, RNAseq data processing and data transfer. We are grateful for funding from HMS through the Dean's Innovation grant. Finally, we are grateful to Shangqin Guo and Amaleah Hartman for the stimulating discussions and thoughtful critique of this paper.

## Additional information

### Funding

| Funder | Grant reference number | Author |
|---|---|---|
| Harvard Medical School | Dean's Innovation Grant | Marc W Kirschner |
| National Institute of General Medical Sciences | GM026875 | Marc W Kirschner |
| National Institute of Child Health and Human Development | HD091846 | Marc W Kirschner |

The funders had no role in study design, data collection and interpretation, or the decision to submit the work for publication.

### Author contributions
Douaa Mugahid, Conceptualization, Data curation, Formal analysis, Investigation, Methodology, Visualization, Writing - review and editing; Marian Kalocsay, Xili Liu, Formal analysis, Methodology, Writing - review and editing; Jonathan Scott Gruver, Methodology, Writing - review and editing; Leonid Peshkin, Formal analysis, Software, Writing - review and editing; Marc W Kirschner, Conceptualization, Funding acquisition, Writing - review and editing

### Author ORCIDs
Douaa Mugahid (iD) https://orcid.org/0000-0002-3455-2992
Leonid Peshkin (iD) https://orcid.org/0000-0002-6420-848X
Marc W Kirschner (iD) https://orcid.org/0000-0001-6540-6130

### Decision letter and Author response
Decision letter https://doi.org/10.7554/eLife.53404.sa1
Author response https://doi.org/10.7554/eLife.53404.sa2

## Additional files

### Supplementary files
- Supplementary file 1. Table with protein and mRNA changes.
- Transparent reporting form

### Data availability
Sequencing data has been deposited in GEO under accession code GSE123296.

The following dataset was generated:

| Author(s) | Year | Dataset title | Dataset URL | Database and Identifier |
|---|---|---|---|---|
| Douaa Mugahid | 2018 | Induction of YAP5SA-GFP vs nGFP expression in FlipinTrex293 cells | https://www.ncbi.nlm.nih.gov/geo/query/acc.cgi?acc=GSE123296 | NCBI Gene Expression Omnibus, GSE123296 |

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
