## [Decision Letter]

**Acceptance summary:**

The Hippo signaling pathway regulates organ size by controlling the activity of the transcription factor, YAP (Yorkie in *Drosophila*). Despite its well-established role in organ size regulation, how YAP coordinately regulates cell number and cell size remains unclear. In this study, the authors dissected the effect of YAP on cell proliferation, cell death and cell size by carefully measuring the growth of cultured mammalian cells at different densities. They show that YAP can regulate these parameters through independent circuits. While YAP stimulates cell proliferation in a cell-autonomous manner, it increases cell size and cell survival largely through diffusible, non-cell autonomous factors. In particular, their studies implicate CYR61, a canonical YAP target gene, as a YAP-induced secreted factor that limits apoptosis in high density cultures, while another unidentified secreted factor likely mediates YAP-dependent increase in cell size. Overall, this study provides a clearer understanding of YAP's role in regulating cell growth on a cellular as well as population level, and highlights the importance of non-cell autonomous signaling downstream of YAP.

**Decision letter after peer review:**

[Editors’ note: the authors submitted for reconsideration following the decision after peer review. What follows is the decision letter after the first round of review.]

Thank you for submitting your work entitled "YAP independently regulates cell size and population growth dynamics via non-cell autonomous mediators" for consideration by *eLife*. Your article has been reviewed by three peer reviewers, one of whom is a member of our Board of Reviewing Editors, and the evaluation has been overseen by a Senior Editor. The reviewers have opted to remain anonymous.

Our decision has been reached after consultation between the reviewers. Based on these discussions and the individual reviews below, we regret to inform you that your work will not be considered further for publication in *eLife*. While the non-autonomous effect of YAP on cell size and cell number is potentially interesting, the reviewers have raised significant technical concerns, detailed below, that have dampened their enthusiasm.

*Reviewer #1:*

YAP is a potent oncogene and is known to regulate both cell growth and proliferation. However, the underlying circuitry by which YAP impacts cell size and cell number is not well understood. In this study, Mugahid, et al. report the interesting finding that in cultured cells, YAP can regulate cell size and cell number independently. They further implicate CYR61, a known secreted YAP target, as the major non-cell autonomous regulator of cell number. These findings are potentially interesting. However, I hope the authors can address the following points.

1) All the analyses presented in this paper have been based on a single cell line. This raises the question of whether the observations are unique to HEK293 cells. Since the authors imply their findings as a general mechanism, it is important to conduct similar analyses in at least one additional cell line.

2) The authors used blocking antibodies to examine the relative contribution of CYR61 versus CTGF in mediating the non-autonomous effect of YAP5SA. Have these antibodies been validated for their blocking activities? I cannot find this information in the main text or the Materials and methods section.

3) At times I find the paper difficult to follow logically. For example, early in the paper the authors stated that the cell size effect is best assessed at low cell density, leading me to assume that all follow-up analyses of cell size would be carried out at low cell density. However, some of the later studies appear to be done using high cell density (e.g., the co-culture experiment). To avoid confusions, it would be helpful for the authors to be explicit about how and why each experiment is done as the way it was described. Also, many figure legends are written in the style of conclusions, with little information on how the data are generated. Overall, I feel that the presentation can be improved with better rationales and interpretations.

*Reviewer #2:*

The authors investigate whether Hippo signaling affects organ size regulation by increasing cell size, cell proliferation, or a combination of both. In particular, the authors argue that YAP activation increases both cell size as well as total cell number at contact inhibition. They further find that the effect of YAP activation acts non-autonomously via a secreted factor CYR61. Since CYR61 has little effect on cell size, they conclude that YAP's effect on cell size and cell number act through separable mechanisms. This paper is potentially interesting, but I have serious technical concerns that significantly dampen my enthusiasm.

There is no indication that the sample size numbers are biological replicates. Much of the reported data appear to be averages of the same experiment conducted on the same clone. Sample sizes should indicate biological replicates not technical replicates (n = 1). This is of critical importance when measuring the cell size of clonal lines. For example, my lab has observed individual clonal lines exhibiting different size homeostasis despite being derived from the same parental line. Therefore, the size increase of nGFP expressing cells compared to WT cells could be due to clonal abnormalities. These experiments should be replicated using multiple clones, and statistically compared to WT cells without co-culture.

A major unaddressed technical problem is the variability of SE dye as well as nuclear area as quantitative measurements of cell size. In our hands, the SE signal depends on the density of cell suspensions at the time of reaction. Cells from the same population fixed and diluted to dense or sparse suspensions and reacted with the same molarity of SE dye yield dramatically different dye signals (denser suspension having lower dye signal). Though it has been shown that within a single sample preparation, SE signal correlate with cell size, it is not generalizable across samples. Therefore, it is crucial for the proper interpretation the size phenotypes that appropriate SE dye calibrations are done to show that the preparation-dependent effects do not confound the results. Similarly, while it has been shown for many cell types that nuclear area linearly scales with overall cell volume, it is unclear how the increase in cell density as contact inhibition occurs may change 3D nuclear morphology and therefore the nuclear area v. cell volume linear relationship. If this linear scaling factor changes during contact inhibition, nuclear area cannot be compared across the growth curve. Especially since the cell size effects under discussion are subtle (<30% change), more detailed controls are needed.

At many points, the figure legends are insufficient for understanding what experiments were performed. The manuscript does not indicate the conditions in which cells were collected for flow analysis (e.g. SE-dye stains). A major point of Figure 1 is that cell size is dependent on culture density, but the density of the cultures used is not explicitly denoted throughout Figures 2-5 or anywhere in the text or Materials and methods.

The authors show no relationship between YAP overactivity and k (Figure 2E), but do show a relationship between CYR61 neutralization and k (Figure 4F). The apparent contradiction between YAP not being necessary for regulating growth rate and CYR61 being sufficient to alter growth rate is not addressed in the manuscript and could be related to the technical concerns listed above.

The main experiments the authors use to argue for non-cell autonomous effects of YAP is via co-culture and neutralizing antibody experiments. Firstly, co-cultures results can be confounded by complex mechanical constraints, especially if the two cell types have different cell sizes, as shown in Figure 2. Secondly, the authors did not show the effect of the anti-CYR61 antibody on wild-type cells (Figure 4A, D), and therefore it is difficult to interpret whether its effects are specific. It would be important to show these controls. Lastly, a complementary experiment showing that conditioned media produced by YAP5A cells can lead to similar effect in increasing k or Ymax in recipient wild-type cells would argue that phenotypes observed in co-culture are indeed mediated through secreted ligands.

*Reviewer #3:*

The manuscript aims to provide insight into the role of YAP transcription factor in determining the size of organs. It reports the results of in vitro experiments with cultured HEK293 cells, which includes the identification of a certain secreted factor (a previously known target of YAP) as a mediator of cell proliferation, controlling the limiting density of a confluent cell culture. Other conclusions and interpretation are however quite equivocal. Overall, in my opinion, this manuscript is substantially below the standard of *eLife*.

The manuscript aims to disentangle regulation of cell size and cell number, but lacks quantitation to do it in a convincing way. Figure 1A-D suggests a simple picture: cell area is constant while cells proliferate before reaching confluence, but goes down after achieving confluence. Rate of proliferation goes to zero as cells reach minimal area. The authors use a logistic equation to describe growth and the resulting cell number with two parameters: one for the rate of growth and the other for the carrying capacity corresponding to the maximal cell density which in general depend on growth conditions and substrate. This is all reasonable. However, curiously, the manuscript fails to note that minimal cell area and the terminal density of cells (and hence the Ymax or "carrying capacity") are directly related (by inverse proportionality). Instead they write: "Since size at the later time points seems to be a complex product of initial size and the achieved Ymax, we focused on changes in nuclear area (cell size) in low density cultures."

There are two major points of confusion here. First: since in the confluent layer cell area and density are inversely proportional to each other, it is not clear what the authors mean by independent control of cell size and cell number. Second: physiology of non-confluent cells is typically quite different from the confluent ones: e.g. mesenchymal/epithelial, so one should not be surprised that factors controlling cell size at low density and the terminal density in confluency may be quite different. This muddies the water as far as the overarching issue of organ size control is concerned, since the paper strongly implies that cell size and the terminal density provide different channels of control. [[See for example the Abstract: "…is the major regulator of the non-cell autonomous increase in cell number, but does not affect cell size. The molecular identity of the non-cell autonomously acting mediator of cell size is yet to be identified."]

I suspect that confusion arises because literature on organ size determination often distinguishes between cell size control (equivalent to cell density) and cell number control with the latter referring to the total cell count in a variable size organ (such as *Drosophila* wing imaginal disc). In that case, cell area and total cell number are independent variables that together define the total area (or "size") of the organ. In contrast, since the in vitro well size is fixed, "Nuclei/Well" measurement performed by the authors report only the density. In the case of a fixed well area, cell size and number are not independent. The fundamental aspect of organ size determination – its global nature – is completely overlooked by the authors. Lacking confinement of the dish well, how do real organs know their size? This work doesn't give us a clue. The confusion involving cell size/cell density/cell number clouds the interpretation of the results and detracts also from the factual findings regarding YAP/secreted factors reported in the paper.

Section V focuses on the effect of substrate stiffness, which is known to affect the growth (and differentiation) of cell growing on them, both at low and high density. It is not surprising that substrate modulates all of the effects reported in the manuscript. What does this teach us about size determination? Relating the findings to the broader thought on role of mechanics in size determination, and what one should and should not expect to see with in vitro adherent cell cultures could avoid confusion. It doesn't help that the authors appear to be unaware of a number of studies addressing the role of mechanical interactions in organ size determination (in *Drosophila*) as well as contact inhibition (in the dish).

The Discussion section revisits 50 years of "contact inhibition" history while missing key ideas (and quantitative studies) on density dependent mobility and the effects of confinement, which would have helped with the logic of the study. Another area of study that is relevant, but overlooked, is the question of "cell competition" which is relevant both to contact inhibition and size determination. Secreted factors have long been suggested as a readouts of tissue mass. How does the secreted factor controlled by YAP relate to those ideas?

In addition to the overall logic of the work I have trouble with some important technical aspects.

1) Logistic fits shown in Figure 1—figure supplement 3C are quite poor at estimating saturation density Ymax. Since measurement of the latter is central to the analysis, I have very little confidence in its accuracy and validity. Furthermore, logistic description at best, kicks in upon confluence, which is reached at different times under different conditions (e.g. it should depend on cell size in the low density state), so an additional fitting parameter would seem to be in order. In either case, since the authors use cell size in a pre-confluent state, one would like to see a more careful/quantitative study of growth rate as a function of cell density (and cell size) in that pre-confluent phase of growth.

2) Do Figures 4A, D 5A, E; 6B, G present data or some sort of a fit (logistic?) to data. Error bars? (Error bars are also missing Figure 1—figure supplement 3. The Materials and methods section does not explain the method used for determining Nuclei/well measurement. Does it involve image segmentation? What are the errors? Could there be a systematic error increasing with increasing cell density?

3) The authors demonstrate the non-autonomous effect of YAP signaling by examining the dependence of Ymax on the fraction of YAP5SA cells in co-culture. Since the effectors are secreted/soluble factors, I would expect that their effect depends on the volume of the media. Does it?

---

## [Author Response]

[Editors’ note: what follows is the authors’ response to the first round of review.]

[…] While the non-autonomous effect of YAP on cell size and cell number is potentially interesting, the reviewers have raised significant technical concerns, detailed below, that have dampened their enthusiasm.Reviewer #1:YAP is a potent oncogene and is known to regulate both cell growth and proliferation. However, the underlying circuitry by which YAP impacts cell size and cell number is not well understood. In this study, Mugahid, et al. report the interesting finding that in cultured cells, YAP can regulate cell size and cell number independently. They further implicate CYR61, a known secreted YAP target, as the major non-cell autonomous regulator of cell number. These findings are potentially interesting. However, I hope the authors can address the following points.1) All the analyses presented in this paper have been based on a single cell line. This raises the question of whether the observations are unique to HEK293 cells. Since the authors imply their findings as a general mechanism, it is important to conduct similar analyses in at least one additional cell line.

We have not established other isogenic cell lines to redo the analyses in their entirety. However, arguably the most important part of the study is to show that the non-autonomous response to YAP is not a special feature of HEK293 cells. We now show that 3T3 cells respond to YAP-induced secreted molecules by increasing their size and number when treated with YAP5SA-conditioned medium (Figure 2—figure supplement 1).

2) The authors used blocking antibodies to examine the relative contribution of CYR61 versus CTGF in mediating the non-autonomous effect of YAP5SA. Have these antibodies been validated for their blocking activities? I cannot find this information in the main text or the Materials and methods section.

The reviewer’s point is well-taken, as we had not included any such data in the original manuscript. We have now included supplementary data (Figure 5—figure supplement 1) which validate that: (1) recombinant CYR61 and CTGF are recognized by the respective antibodies and that (2) the same antibodies recognize the proteolytically cleaved (active) forms of CYR61 and CTGF in concentrates of conditioned medium from YAP5SA and nGFP-expressing cells. The data only shows that the antibody recognizes the correct protein but does not prove inactivation per say. As there seems to be no well-established assay for CYR61 activity at present, it was not possible to show direct inactivation. However, we do find the same results using a second antibody raised against a different epitope of the CYR61 protein making it unlikely that it’s simply an off-target effect (Figure 5—figure supplement 1I, HJ). We are further reassured that the results are CYR61- specific as addition of purified recombinant CYR61 to cells results in changes opposite to those seen with antibody treatment.

3) At times I find the paper difficult to follow logically. For example, early in the paper the authors stated that the cell size effect is best assessed at low cell density, leading me to assume that all follow-up analyses of cell size would be carried out at low cell density. However, some of the later studies appear to be done using high cell density (e.g., the co-culture experiment). To avoid confusions, it would be helpful for the authors to be explicit about how and why each experiment is done as the way it was described. Also, many figure legends are written in the style of conclusions, with little information on how the data are generated. Overall, I feel that the presentation can be improved with better rationales and interpretations.

We have included a more detailed explanation of each specific experiment both in the main text and in the figure legends. We realize that this was a general concern all reviewers shared. Regarding the particular comment on cell size measurements and density, most experiments are done at low density as mentioned in the main text, but we had mentioned there would be some exceptions. As a minimum number of cells is needed to effectively condition the culture medium over a certain course of time, co- cultures were an instance of the exception to the rule. In that case, the total number of cells was not just YAP5SA cells, and we observed that the non-cell autonomous effects on size were more prominent after longer culture times at which cell density is often higher than what we consider low density. We attribute that to the need for a longer time/higher number of cells before the soluble mediator accumulates to effective levels.

Reviewer #2:The authors investigate whether Hippo signaling affects organ size regulation by increasing cell size, cell proliferation, or a combination of both. In particular, the authors argue that YAP activation increases both cell size as well as total cell number at contact inhibition. They further find that the effect of YAP activation acts non-autonomously via a secreted factor CYR61. Since CYR61 has little effect on cell size, they conclude that YAP's effect on cell size and cell number act through separable mechanisms. This paper is potentially interesting, but I have serious technical concerns that significantly dampen my enthusiasm.There is no indication that the sample size numbers are biological replicates.

We assure the reviewer that the sample sizes reflect biological replicates when stated because these measurements were done on several independent live cell cultures in parallel. On the Incucyte we can measure the number of nuclei and their average area across multiple wells, and what we state as a biological replicate are the number of wells compared per group. We have always repeated the entire set of experiments at least once to confirm the robustness of the changes. In case of the SE-dye data which we reported in the first submission, we depicted histograms from a single population per condition and did not report any statistics. The data were a representative sample of experiments repeated at least twice using the same method and showed similar qualitative changes. For the QPM measurements, which we have now included and base our findings on, the replicates come from 3 different, yet isogenic clones.

We have clarified what n represent in the respective figure legends.

Much of the reported data appear to be averages of the same experiment conducted on the same clone. Sample sizes should indicate biological replicates not technical replicates (n = 1). This is of critical importance when measuring the cell size of clonal lines. For example, my lab has observed individual clonal lines exhibiting different size homeostasis despite being derived from the same parental line. Therefore, the size increase of nGFP expressing cells compared to WT cells could be due to clonal abnormalities. These experiments should be replicated using multiple clones, and statistically compared to WT cells without co-culture.

a) We of course agree. We addressed the issue of replicates in the prior point. However, the reviewer does raise a valid point on clonal differences, which are somewhat different from the issue of biological replicates. Having considered that issue ourselves, we made sure we used FlipinTrex-293 cells which allow the generation of isogenic, Dox-inducible cell lines for the bulk of our most informative experiments as indicated in the Materials and methods section. These are the cells we used for making comparisons between the 5SA vs. nGFP cell lines and anything depending on them. This eliminates the problem of the variable site of genomic integration of the plasmid in the different clones, which, as the reviewer points out, could make the comparisons less reliable if the cell lines were not isogenic. We have now explained the rationale for that choice more thoroughly in the main text and not just the Materials and methods. To further confirm the soundness of our choice, and because we take the reviewer’s concerns quite seriously considering the importance of these findings, we went back and measured the dry mass of 3 different 5SA and GFP clones and compared them to the parental population using quantitative phase microscopy (Figures 2A and Figure 1—figure supplement 5D). These measurements demonstrate that:i) The average dry mass of the different nGFP clones is 283.6 ± 12.3 pg, while the average of the 5SA clones is 316.1 ± 6.3 under the conditions we typically keep them.

i) The difference between nGFP-expressing clones and the parental population is 1.5% , and suggests that the ~ 10% difference observed using the protein dye measurements did not reflect true biological differences.

ii) The 5SA-expressing clones are ~20% larger than nGFP clones in the presence of Dox through all the phases of the cell cycle, which is statistically significant (p<0.001, two-way ANOVA) and represents a difference of 5 standard deviations.

iii) The 5SA-expressing clones are significantly larger than nGFP clones through all the phases of the cell cycle (bet. 4-9%, *p* < 0.01, two-way ANOVA) even in the absence of Dox, which we attribute to the production of a soluble mediator of cell size by the fraction of leaky cells in the 5SA cultures.

b) We do want to note that the HEK293 cells we used for the comparison between GFP-WTYAP vs. nGFP cell in **Figure 1F**and the YAPsh vs. empty vector-expressing lines are not isogenic. Considering that the WT-YAP experiment is only a validation of the FlipinTrex data, we feel comfortable including the data. It could also serve as an example of an experiment on an independent clone. We are less concerned about the YAPsh cells and their controls, as in this case we are reducing protein expression. In summary concern with clonal differences does not apply to our key experiments.

c) The reviewer requested that we include representative data on size distribution from non-co- cultured nGFP and 5SA cells. We have now included that data as the population mean ± S.D. (**Figure 1G**). We also refer the reviewers’ attention to the data in Figure 1—figure supplement 5Dwhich depicts the mean dry mass of three different clones per condition and the S.D.

A major unaddressed technical problem is the variability of SE dye as well as nuclear area as quantitative measurements of cell size. In our hands, the SE signal depends on the density of cell suspensions at the time of reaction. Cells from the same population fixed and diluted to dense or sparse suspensions and reacted with the same molarity of SE dye yield dramatically different dye signals (denser suspension having lower dye signal). Though it has been shown that within a single sample preparation, SE signal correlate with cell size, it is not generalizable across samples. Therefore, it is crucial for the proper interpretation the size phenotypes that appropriate SE dye calibrations are done to show that the preparation-dependent effects do not confound the results. Similarly, while it has been shown for many cell types that nuclear area linearly scales with overall cell volume, it is unclear how the increase in cell density as contact inhibition occurs may change 3D nuclear morphology and therefore the nuclear area v. cell volume linear relationship. If this linear scaling factor changes during contact inhibition, nuclear area cannot be compared across the growth curve. Especially since the cell size effects under discussion are subtle (<30% change), more detailed controls are needed.

a) In retrospect this is the point that I am happiest that the reviewer forced us to reconsider. Our original data were inadequate and it took very different experiments to enforce the points we wished to make. The concern of the reviewer went to the inherent reliability of the dye measurement, which was justifiable. The dye can be quantitative but it is not an absolute measurement thus raising doubts about its validity across samples. Furthermore, it is not very precise (Cv of ~20%) Though we agree that it may be possible to find ways to make comparisons reliable if the experiment is set up carefully, it will always be suspect in the absence of internal controls. In fact, that is one of the main reasons we used nuclear area as a proxy for cell size through much of the paper, as it allows the comparison of replicates, and conditions over time with little worry about technical variability due to the staining process, etc. That said, in doing the SE-staining, all conditions represented in one figure were processed as identically as possible, meaning they were all seeded on the same day, trypsinized, fixed and permeabilized simultaneously at the same count of cells per experiment (1-2 x106 cell/ml), and stained and analyzed at the same time. Our conclusions are based on the reproducibility of the trends across at least 3 independent repeats showing the same qualitative differences (for example Figures 1E, Figure 1—figure supplement 6Bare independent experiments done in single cell type cultures). We used the Coulter counter, which measures yet another metric of cell size: cell volume. The data are included as Figure 1—figure supplement 5A in the new submission and shows that there is a ~ 25% increase in cell volume in cells expressing YAP5SA. In this method no sample processing is required post-trypsinization and cell count should have no effect on the readout. A limitation here is that volume may be less significant than dry mass in that volume change could be due to uptake of water. Furthermore, as is the case with nuclear area it does not allow us to differentiate between cells at different phases of the cell cycle. Therefore, we used a second method based on interferometry, Quantitative Phase Microscopy (QPM), which we have recently improved by various computational means. QPM has a Cv of 2%. We measured dry cell mass of 3 clones of the two cell lines in their attached form as mentioned in response to the prior point. Using Hoechst to stain the nuclei of live cells, we were able to bin the cells into their respective cell cycle phase. The QPM data shows that 5SA cells are ~ 20% greater than nGFP controls throughout the cell cycle (Figure 1—figure supplement 5D).

Please note how tight the distributions are with the SD in all cases equal to, at most, 2% of the mean.

b) We also agree with the reviewer that we cannot assume that scaling between cell size and nuclear area in high density cultures would be the same as at low density. To address the validity of nuclear area as a proxy for cell size we used QPM to measure the dry mass per cell and compared it to the cell’s nuclear area. We know that for this cell type, nuclear area and dry mass are highly correlated across the different measurements we have done (Pearson correlation coefficient ~0.9; **Figure 1—figure supplement 2A**). As we can only reliably quantify dry cell mass in low density cultures (as cell-cell adhesion interferes with accurate segmentation), we could not obtain such high-quality data in confluent cultures. We certainly do not claim that the relationship between cell size and nuclear area necessarily scales 1:1 across the entirety of the growth curve, i.e.: there might not be an exponential decrease in cell size in high density cultures as reflected by the nuclear area. Nevertheless we do see a decrease in cell size at high density as reflected in: (i) a decrease in protein content using the SE dye (**Figures 1F, Figure 1—figure supplement 5E, G**), (ii) a decrease in protein content as reflected by BCA protein quantification from the same number of cells from high, medium and low density cultures (**Figure 1—figure supplement 4D**) and (iii) a decrease in cell volume measured by Coulter Counter as shown in **Figure 1—figure supplement 4C**. With that in mind, we have focused most of our work in the later parts of the manuscript (beyond **Figure 1**) on measurements in sub-confluent cultures, where QPM gives very accurate information about the dry mass of the cell.

At many points, the figure legends are insufficient for understanding what experiments were performed. The manuscript does not indicate the conditions in which cells were collected for flow analysis (e.g. SE-dye stains). A major point of Figure 1 is that cell size is dependent on culture density, but the density of the cultures used is not explicitly denoted throughout Figures 2-5 or anywhere in the text or Materials and methods.

We have revisited the legends for clarification as suggested by several of the reviewers. Cell number, the culture vessel and the approximate corresponding density have been explicitly reported in each legend.

The authors show no relationship between YAP overactivity and k (Figure 2E), but do show a relationship between CYR61 neutralization and k (Figure 4F). The apparent contradiction between YAP not being necessary for regulating growth rate and CYR61 being sufficient to alter growth rate is not addressed in the manuscript and could be related to the technical concerns listed above.

We would like to point out that the apparent “discrepancy” between the effect of CYR61 and k has been confirmed in several independent experiments. More specifically we consistently see that (1) the addition of recombinant CYR61 decreases k (Figure 5G, J), (2) the addition of anti-CYR61 antibody increases k Figure 5A, D), while (3) YAP5SA expression increases k (Figure 1D) and (4) knocking down YAP decreases k (Figure 3E). Our new data also supports that (1) YAP expression increases Ymax in a CYR61-dependent manner, but (2) YAP’s strongest effects on k are cell autonomous. Given that YAP mediates the transcription of not only CYR61, which is secreted, but a multitude of intracellular proteins, it is not completely surprising that the effect of YAP on k is not CYR61-dependent. One possible explanation for seeing an effect on k at all when low- density cells (still not contact inhibited) are treated with recombinant CYR61 or anti-CYR61 antibodies is that the effective concentration of CYR61 in these cultures becomes less dependent of the amount accumulated in the medium due to cellular production. We have not exhaustively investigated all other possible explanations, but our data does lead us to conclude with relative confidence that CYR61 and YAP overexpression do not have the same effect on k. We have added our thoughts on this discrepancy to the manuscript.

The main experiments the authors use to argue for non-cell autonomous effects of YAP is via co-culture and neutralizing antibody experiments. Firstly, co-cultures results can be confounded by complex mechanical constraints, especially if the two cell types have different cell sizes, as shown in Figure 2. Secondly, the authors did not show the effect of the anti-CYR61 antibody on wild-type cells (Figure 4A, D), and therefore it is difficult to interpret whether its effects are specific. It would be important to show these controls. Lastly, a complementary experiment showing that conditioned media produced by YAP5A cells can lead to similar effect in increasing k or Ymax in recipient wild-type cells would argue that phenotypes observed in co-culture are indeed mediated through secreted ligands.

a) We agree that co-cultures can be confounded by mechanical constraints. It is something that we specifically considered, and it is not clear why this was not commented on. We did not stop with co- culture but used a Transwell assay where the cells are not in physical contact (**Figure 2D, E**). We further showed the same activity using medium conditioned by 5SA-expressing cells (**Figure 3H**), where there are no possible interactions between living cells, such as reciprocal signaling.

b) We have included a supplementary figure on the effect of CYR61 on pure mCherry-expressing WT/nGFP cells (Figure 2—figure supplement 1G, J), which indicate that neutralizing CYR61 in WT cells also decreases Ymax. Furthermore, the new data in **Figure 6** is based on data from those WT cells which produce 8 times less CYR61 than YAP5SA cells. We also do not see any titratable effects on Ymax when cells are treated with a control polyclonal Rabbit IgG antibody.

c) The issue of specificity has been addressed in more detail in the response to point 2 of reviewer 1’s comments.

Reviewer #3:The manuscript aims to provide insight into the role of YAP transcription factor in determining the size of organs. It reports the results of in vitro experiments with cultured HEK293 cells, which includes the identification of a certain secreted factor (a previously known target of YAP) as a mediator of cell proliferation, controlling the limiting density of a confluent cell culture. Other conclusions and interpretation are however quite equivocal. Overall, in my opinion, this manuscript is substantially below the standard of eLife.The manuscript aims to disentangle regulation of cell size and cell number, but lacks quantitation to do it in a convincing way. Figure 1A-D suggests a simple picture: cell area is constant while cells proliferate before reaching confluence, but goes down after achieving confluence. Rate of proliferation goes to zero as cells reach minimal area. The authors use a logistic equation to describe growth and the resulting cell number with two parameters: one for the rate of growth and the other for the carrying capacity corresponding to the maximal cell density which in general depend on growth conditions and substrate. This is all reasonable. However, curiously, the manuscript fails to note that minimal cell area and the terminal density of cells (and hence the Ymax or "carrying capacity") are directly related (by inverse proportionality). Instead they write: "Since size at the later time points seems to be a complex product of initial size and the achieved Ymax, we focused on changes in nuclear area (cell size) in low density cultures."There are two major points of confusion here. First: since in the confluent layer cell area and density are inversely proportional to each other, it is not clear what the authors mean by independent control of cell size and cell number. Second: physiology of non-confluent cells is typically quite different from the confluent ones: e.g. mesenchymal/epithelial, so one should not be surprised that factors controlling cell size at low density and the terminal density in confluency may be quite different. This muddies the water as far as the overarching issue of organ size control is concerned, since the paper strongly implies that cell size and the terminal density provide different channels of control. [[See for example the Abstract: "…is the major regulator of the non-cell autonomous increase in cell number, but does not affect cell size. The molecular identity of the non-cell autonomously acting mediator of cell size is yet to be identified."]I suspect that confusion arises because literature on organ size determination often distinguishes between cell size control (equivalent to cell density) and cell number control with the latter referring to the total cell count in a variable size organ (such as *Drosophila* wing imaginal disc). In that case, cell area and total cell number are independent variables that together define the total area (or "size") of the organ. In contrast, since the in vitro well size is fixed, "Nuclei/Well" measurement performed by the authors report only the density. In the case of a fixed well area, cell size and number are not independent. The fundamental aspect of organ size determination – its global nature – is completely overlooked by the authors. Lacking confinement of the dish well, how do real organs know their size? This work doesn't give us a clue. The confusion involving cell size/cell density/cell number clouds the interpretation of the results and detracts also from the factual findings regarding YAP/secreted factors reported in the paper.

We of course agree with the reviewer that organ size control (with the potentially unlimited boundaries for growth), and the growth of a population on a dish are potentially determined by different factors. We now make clearer that this paper is *not* about organ size control, and the tissue culture system is not being used to draw one-to-one parallels to organ size regulation in vivo. That said, cells proliferate and die, and cells have a mean size. These can and do vary independently physiologically and upon perturbation. We have simply used the tissue culture system to learn about the effect of YAP on (1) cell size, (2) cell division and (3) death rates. As the balance of all three parameters affects organ growth, the findings could be relevant to how YAP regulates organ size but drawing any specific conclusion about organ size goes beyond our experiments. Our interpretation of our results in the context of organ size is at most speculative and it is up to future experiments to show how these contribute to growth and size regulation of organs. It is not clear to us whether the reviewer believes that we are confused (we are not) or that we might confuse others. We have cleared this up in the Discussion where we discuss more precisely what the relevance of our findings in tissue culture could be the control of organ size in what we hope is a non- dogmatic way. We have also differentiated between the in vivoand culture context in the Introduction for clarity.

Second, we would argue that while the area of a dish is finite, a higher number of cells does not necessarily mean cells need to become smaller. The area of the well limits the substrate-attached area of the cells, but cells can continue to grow in height. So, fitting more cells does not necessarily require them to be smaller in size. As the reviewer has speculated, it is tempting to think that cells stop proliferating simply because they have reached a minimum size, and as YAP5SA cells are larger initially they can proliferate more times before they stop dividing. However, that did not turn out to be the case, and this was not the independence referred to in the title. The “independence” in the title refers to what turned out to be the effect of CYR61 on the number of cells at high density by affecting the death rates, without affecting cell size. That independence between proliferation rates and cell size was further emphasized by the finding that the effect of YAP5SA on cell division rates (represented in k) seem to be dominated by cell autonomous effectors, while the size effects are non-cell autonomous. We appreciate that the reviewer might have been especially worried about confusion for others. We have endeavored throughout this paper to point out that we are talking about cell communication and *cellular* phenomena and not organ size while at the same time pointing out that there might be some implications of these studies in an in vivocircumstance.

The Discussion section revisits 50 years of "contact inhibition" history while missing key ideas (and quantitative studies) on density dependent mobility and the effects of confinement, which would have helped with the logic of the study. Another area of study that is relevant, but overlooked, is the question of "cell competition" which is relevant both to contact inhibition and size determination. Secreted factors have long been suggested as a readouts of tissue mass. How does the secreted factor controlled by YAP relate to those ideas?

Cell-cell competition is quite an interesting idea to explore, and we did not overlook it on purpose, but believe that our data does not provide more than loose speculation on it. As the new version of the manuscript also confirms that the effects we observe in co-culture can be mediated by conditioned medium, we believe competition might be playing a smaller role with respect to the parameters we studied here, but of course does not rule out its overall importance in other contexts. As to the role of secreted proteins in organ size regulation in vivo, etc., we do speculate broadly on that in the last paragraph or so of the Discussion but tried to keep the breadth of our speculations within reasonable length and within the immediate implication of the data we have.

In addition to the overall logic of the work I have trouble with some important technical aspects.1) Logistic fits shown in Figure 1—figure supplement 3C are quite poor at estimating saturation density Ymax. Since measurement of the latter is central to the analysis, I have very little confidence in its accuracy and validity. Furthermore, logistic description at best, kicks in upon confluence, which is reached at different times under different conditions (e.g. it should depend on cell size in the low density state), so an additional fitting parameter would seem to be in order. In either case, since the authors use cell size in a pre-confluent state, one would like to see a more careful/quantitative study of growth rate as a function of cell density (and cell size) in that pre-confluent phase of growth.

While the reviewer might not find the data in Figure 1—figure supplement 3C) convincing (R2>0.99), the main explanation for the sup-exemplary fit is that we do not have enough data points in the later phases of the growth curve. We have included all the new data with the original and fitted data, all of which have more points in the plateau phase of growth and show a near-perfect fit (see **Figures 1C, 2E, 3E, Figure 1—figure supplement 5G**). We observe that 5SA cells, although larger in size in the low-density regime, start to decrease in size later than the WT cells. So if they were reaching confluence earlier, their response to contact-inhibition is mildly retarded, which wouldn’t be captured by the simple additional parameter the reviewer requests be added to a basic logistic model.

That being said, we have taken the reviewer’s comment seriously and used a model-agnostic method of estimating Ymax. In this case Ymax is simply the number of nuclei/well at which population growth plateaus (rate of change is zero; termed Ymax’). We estimate the rate of growth (k’) by fitting only the earlier time points for each data set to an exponential growth equation. Unsurprisingly, this analysis yields qualitatively similar results as k in the logistic function approximates to the exponential growth rate at low density by definition. The changes in maximum cell number (Ymax’) are also maintained. As Ymax’ and k’ do not change the conclusions of the manuscript in any way, we continue to use the logistic fit in the main figures.

2) Do Figures 4A, D 5A, E; 6B, G present data or some sort of a fit (logistic?) to data. Error bars? (Error bars are also missing Figure 1—figure supplement 3. The Materials and methods section does not explain the method used for determining Nuclei/well measurement. Does it involve image segmentation? What are the errors? Could there be a systematic error increasing with increasing cell density?

We had originally excluded error bars because we are actually just showing the fitted data. We have made a point of including the raw data with error bars and their fit in all new panels.

We mention in the Materials and methods section that nuclei were segmented using the image analysis software on the Incucyte server. While there might be an increase in segmentation error at higher density, it is off-set by an increase in the total number of nuclei per well. Overall, segmenting nuclei is relatively robust on the various platforms we have tried. We have also validated that there is a decrease in cell size with density using a coulter counter (cell volume), BCA estimation of total protein content from the same number of cells from cultures of different densities, as well as the SE-dye (reflecting total protein content; See response to reviewer 2).

3) The authors demonstrate the non-autonomous effect of YAP signaling by examining the dependence of Ymax on the fraction of YAP5SA cells in co-culture. Since the effectors are secreted/soluble factors, I would expect that their effect depends on the volume of the media. Does it?

We certainly see a decrease in the effect of YAP5SA-conditioned medium on cell number when diluted. See Author response image 1.

**Author response image 1. respfig1:** 1 pt lines reflect the mean and 0.25 pt lines are ± SEM.